# Locally Optimal Fixed-Budget Best Arm Identification in Two-Armed Gaussian Bandits with Unknown Variances

## Abstract

We address the problem of best arm identification (BAI) with a fixed budget for two-armed Gaussian bandits. In BAI, given multiple arms, we aim to find the best arm, an arm with the highest expected reward, through an adaptive experiment. Kaufmann et al. (2016) develops a lower bound for the probability of misidentifying the best arm. They also propose a strategy, assuming that the variances of rewards are known, and show that it is asymptotically optimal in the sense that its probability of misidentification matches the lower bound as the budget approaches infinity. However, an asymptotically optimal strategy is unknown when the variances are unknown. For this open issue, we propose a strategy that estimates variances during an adaptive experiment and draws arms with a ratio of the estimated standard deviations. We refer to this strategy as the *Neyman Allocation (NA)-Augmented Inverse Probability weighting (AIPW)* strategy. We then demonstrate that this strategy is asymptotically optimal by showing that its probability of misidentification matches the lower bound when the budget approaches infinity, and the gap between the expected rewards of two arms approaches zero (*small-gap regime*). Our results suggest that under the worst-case scenario characterized by the small-gap regime, our strategy, which employs estimated variance, is asymptotically optimal even when the variances are unknown.

## 1 Introduction

This study investigates the problem of *best arm identification (BAI) with a fixed budget* in stochastic two-armed Gaussian bandits. In this problem, we consider an adaptive experiment with a fixed number of rounds, called a *budget*. At each round, we can draw an arm and observe the reward. The goal of the problem is to identify the best arm with the highest expected reward at the end of the experiment (Bubeck et al., 2009; Audibert et al., 2010).

Formally, we consider the following adaptive experiment with two arms and Gaussian rewards. There are two arms 1 and 2, and an arm $a \in \{1, 2\}$ has an $\mathbb{R}$-valued Gaussian reward $Y_a \sim \mathcal{N}(\mu_a, \sigma_a^2)$ with the mean $\mu_a \in [-C_\mu, C_\mu]$ and the variance $\sigma_a^2 \in [C_{\sigma^2}, 1/C_{\sigma^2}]$ for some universal constants $C_\mu, C_{\sigma^2} > 0$. We assume that $C_\mu$ and $C_{\sigma^2}$ are *known* to us for a technical purpose, and it is enough to set $C_\mu$ as a sufficiently large value and $C_{\sigma^2}$ as a sufficiently small value. Given fixed $(\sigma_1^2, \sigma_2^2)$, let

$$\mathcal{P}^{\mathrm{G}} := \mathcal{P}^{\mathrm{G}}_{(\sigma_1^2, \sigma_2^2)} := \left\{ P = (\mathcal{N}(\mu_1, \sigma_1^2), \mathcal{N}(\mu_2, \sigma_2^2)) : \mu_1, \mu_2 \in (-\infty, +\infty), \ \ \mu_1 \neq \mu_2 \right\}$$

be a set of distributions generating the data, which is referred to as the *Gaussian bandit models*, where $P \in \mathcal{P}$ is a pair of distributions that generate $(Y_1, Y_2)$, and $\mathcal{N}(\mu, \sigma^2)$ is a Gaussian distribution with a mean $\mu$ and a variance $\sigma^2$. For an instance $P$, the best arm $a^\star(P) \in \{1, 2\}$ is defined as $a^\star(P) = \arg\max_{a \in \{1,2\}} \mu_a$, which is assumed to exist uniquely.

In the adaptive experiment, we consider a strategy to identify the best arm. A fixed budget $T$ is given. For each round $t \in [T] := \{1, 2, \ldots, T\}$, let $(Y_{1,t}, Y_{2,t})$ be an independent and identically distributed (i.i.d.) copy of $(Y_1, Y_2)$. At each round $t$, we draw arm $A_t \in \{1, 2\}$ and observe a reward $Y_t = \sum_{a \in \{1,2\}} \mathbb{1}[A_t = a] Y_{a,t}$.

At the end of the experiment (after round $T$), we recommend an estimated best arm $\widehat{a}_T \in \{1, 2\}$. During an experiment, we follow a *strategy* that determines which arm to draw and which arm to recommend as the best arm. The performance of strategies is evaluated by a minimal probability of misidentification $\mathbb{P}_P(\widehat{a}_T \neq a^\star(P))$, where $\mathbb{P}_P$ is the probability law under $P$.

**Background.** In fixed-budget BAI, it has been an important question of interest to investigate the probability of misidentification $\mathbb{P}_P(\widehat{a}_T \neq a^\star(P))$ in the limit $T \to \infty$. For the interest, a typical approach is to derive an upper and lower bound of the probability separately and specify its value.

For a lower bound of the probability of misidentification, Kaufmann et al. (2016) develops a general theory for deriving lower bounds of the probability. Their theory applies the change-of-measure argument, which has been employed in various problems (van der Vaart, 1998), including studies for regret minimization (Lai & Robbins, 1985). Their lower bound is general and can be applied to a wide range of settings, such as the fixed confidence setting (Garivier & Kaufmann, 2016) as well as the fixed budget setting.

In contrast, an upper bound of the misidentification probability has not been fully clarified. A typical way to derive upper bounds is to construct a specific strategy and evaluate its misidentification probability. Kaufmann et al. (2016) develops a strategy under a setting in which the variance $(\sigma_1^2, \sigma_2^2)$ of the reward is known and shows its misidentification probability corresponds to the lower bound. However, this strategy is not available under the usual setting with unknown variance. Based on these situations, the current results are insufficient to establish an upper bound for the misidentification probability when the variances are unknown.

Based on the situation above, our interest is in strategies for identifying misidentification probabilities in the adaptive experimental setting described above. Specifically, we need a strategy such that an upper bound on its misidentification probability aligns with the lower bound proposed in Kaufmann et al. (2016). Further, this strategy must be valid when the variance is unknown.

**Our approach and contribution.** In this study, we develop a strategy whose probability of misidentification aligns with the lower bound under an additional setting. To accomplish this, we develop the *Neyman allocation-augmented inverse probability weight* (NA-AIPW) strategy. Then, we show that the probability of misidentification aligns with the lower bound under a *small-gap regime*. The details of each are described below.

The NA-AIPW strategy consists of a sampling rule using the Neyman allocation (NA) and a recommendation rule using the augmented inverse probability weighting (AIPW) estimator. NA is a method of sampling arms using a ratio of the root of the variance of rewards, as utilized in Neyman (1934); Kaufmann et al. (2016). The NA-AIPW strategy samples the arms by estimating this variance during the adaptive experiment. At the end of the experiment, the NA-AIPW strategy recommends an arm with the highest expected reward estimated by using the AIPW estimator, which is an unbiased estimator with a small asymptotic variance.

The small-gap regime considers a situation $\mu_1 - \mu_2 \to 0$ as $T \to \infty$. Although this additional setting slightly simplifies the problem with BAI, the problem is still sufficiently complicated since the small gap makes it difficult to identify the best arm. This setting has been utilized in BAI with fixed confidence, such as the analysis of lil'UCB (Jamieson et al., 2014). In the realm of statistical testing, such an evaluation framework is known as the local Bahadur efficiency (Bahadur, 1960; Wieand, 1976; Akritas & Kourouklis, 1988; He & Shao, 1996). From a technical perspective, the small-gap regime is a situation where we can ignore the estimation error of the variances compared to the difficulty of identifying the best arm. Since the error of the estimation of the variance is relatively negligible in the small-gap setting, we can show that the misidentification probability of the NA-AIPW strategy matches the lower bound.

We summarize the backgrounds and our contributions. In BAI with two-armed Gaussian rewards and a fixed budget, a strategy has been needed in which its misidentification probability achieves the lower bound derived by Kaufmann et al. (2016). Although Kaufmann et al. (2016) demonstrates an asymptotically optimal strategy that satisfies the requirement with known variances, it remains an unresolved issue to find a strategy whose upper bound matches their derived lower bound when variances are unknown. For this

issue, this study proposes the NA-AIPW strategy whose probability of misidentification matches the lower bound under the small-gap regime.

**Organization.** This study is organized as follows. First, in Section 2, we review the lower bound of Kaufmann et al. (2016). Then, in Section 3, we propose our NA-AIPW strategy. In Section 4, we show that the misidentification probability of the strategy asymptotically corresponds to the lower bound by Kaufmann et al. (2016) under the small-gap setting. We show the proof in Section 5, where we also provide a novel concentration inequality based on the Chernoff bound. In Section 6, we discuss the difficulty in this problem. In Section 7, we introduce related work and remaining problems, which includes an extension of our small-gap setting to a setting with multi-armed bandits and non-Gaussian rewards.

**Notation**. Let $\mathcal{F}_t$ be the sigma-algebra generated by all observations up to round $t$. We define a truncation operator: for a variable $v \in \mathbb{R}$ and a constant $c \geq 1$, $\mathrm{thre}(v; c_1, c_2) \coloneqq \min\{\max\{v, c_1\}, c_2\}$.

## 2  Lower Bound of Probability of Misidentification

As a preparation, we introduce a lower bound for the probability of misidentification in BAI with a fixed budget. We call a strategy is *consistent*, if for any $P \in \mathcal{P}^{\mathrm{G}}$, $\mathbb{P}_P(\widehat{a}_T \neq a^\star(P)) \to 0$ as $T \to \infty$. To evaluate the performance of strategies For any $P \in \mathcal{P}^{\mathrm{G}}$, we focus on the following metric for $\mathbb{P}_P(\widehat{a}_T \neq a^\star(P))$ used in many studies, such as Kaufmann et al. (2016):

$$-\frac{1}{T} \log \mathbb{P}_P(\widehat{a}_T \neq a^\star(P)).$$

Note that the upper bound (resp. lower bound) of this term works as a lower bound (resp. upper bound) of the probability of misidentification $\mathbb{P}_P(\widehat{a}_T \neq a^\star(P))$ since $x \mapsto -\log x$ is a strictly decreasing function.

For two-armed Gaussian bandits, Kaufmann et al. (2016) presents the following lower bounds.

**Proposition 2.1** (Theorem 12 in Kaufmann et al. (2016))**.** *For any $P^* \in \mathcal{P}^{\mathrm{G}}$ and $\Delta = \mu_1^* - \mu_2^*$ with known constants $C_\mu, C_{\sigma^2} > 0$ independent of $T$, if $\{(Y_{1,t}, Y_{2,t})\}_{t \in [T]}$ is generated from $P^*$, any consistent strategy satisfies*

$$\limsup_{T \to \infty} -\frac{1}{T} \log \mathbb{P}_{P^*}(\widehat{a}_T \neq a^\star(P^*)) \leq \frac{\Delta^2}{2(\sigma_1 + \sigma_2)^2}.$$

Note that we can remove the condition that we know constants $C_\mu, C_{\sigma^2} > 0$ independent of $T$ such that $\mu_a \in [-C_\mu, C_\mu]$ and $\sigma_1^2, \sigma_2^2 > C_{\sigma^2}$ hold for deriving the lower bound. However, it is required to implement our strategy and to derive an upper bound. For the conditions of the lower bound to align with those of the upper bound, we add the conditions in this proposition.

From the statement, there are some important aspects of this lower bound: (i) The term $\Delta = \mu_1^* - \mu_2^*$, which referred to a *gap*, appears in the numerator and the magnitude of the error is described by the gap. (ii) The variances $(\sigma_1^2, \sigma_2^2)$ appear in the denominator, which plays an important role.

It has been discussed to find a strategy in which the upper bound of its probability of misidentification coincides with this lower bound in Proposition 2.1. Although Kaufmann et al. (2016) develops a strategy that satisfies the requirement, it needs to sample arms with some probability depending on the known variances $(\sigma_1^2, \sigma_2^2)$. To the best of our knowledge, if the variances are unknown and need to be estimated during adaptive experiments, no one has found the desired strategy.

## 3  The NA-AIPW Strategy

In this section, we define our strategy. Formally, a strategy gives a pair $((A_t)_{t \in [T]}, \widehat{a}_T)$, where (i) $(A_t)_{t \in [T]} \in \{1, 2\}^T$ is a sequence of arms generated by a sampling rule that determines which arm $A_t$ is chosen in each $t$ based on $\mathcal{F}_{t-1}$, and (ii) $\widehat{a}_T \in \{1, 2\}$ is a recommended arm by a recommendation rule based on $\mathcal{F}_T$. Our proposed NA-AIPW strategy consists of (i) a sampling rule with the Neyman Allocation (NA) (Neyman,

1923), and (ii) a recommendation rule using the Augmented Inverse Probability Weighting (AIPW) estimator (Robins et al., 1994; Bang & Robins, 2005). Based on these rules, we refer to this strategy as the NA-AIPW strategy[1].

## 3.1 Target Allocation Ratio

As preparation, we introduce the notion of a target allocation ratio, which will be used for the sampling rule. We define target allocation ratios $w_1^*, w_2^* \in (0, 1)$ as

$$w_1^* := \frac{\sigma_1}{\sigma_1 + \sigma_2}, \quad \text{and} \quad w_2^* := 1 - w_1^*.$$

A sampling rule following this target allocation ratio is known as the Neyman allocation rule (Neyman, 1934). Glynn & Juneja (2004) and Kaufmann et al. (2016) also propose this allocation. This target allocation ratio is characterized by the variances (standard deviations); therefore, the target allocation ratio is unknown when the variances are unknown. Therefore, to use this ratio, we need to estimate it from observations.

## 3.2 Sampling Rule with Neyman Allocation (NA)

We present the sampling rule with the NA. At each round $t \in [T]$, our sampling rule randomly draws an arm $a \in \{1, 2\}$ with a probability identical to an estimated version of the target allocation ratio $w_a^*$. To estimate the target allocation ratio $w_a^*$, we estimate the variances during the adaptive experiment. For $a \in \{1, 2\}$, let $\{\widehat{\sigma}_a\}_{t \in [T]}$ and $\{\widehat{w}_{a,t}\}$ be sequences of estimators of $\sigma_a, \mu_a$ and $w_a^*$, that will be defined bellow.

We use the rounds $t = 1$ and $t = 2$ for initialization. Specifically, we draw the arm 1 at round $t = 1$ and the arm 2 at round $t = 2$, and also set $\widehat{w}_{a,1} = \widehat{w}_{a,2} = 1/2$ for $a \in \{1, 2\}$.

At the round $t \geq 3$, we estimate the target allocation ratio (variances) $w_a^*$ for $a \in \{1, 2\}$ using past observations $\mathcal{F}_{t-1}$. For each $t \geq 3$, we first define an estimator of the expected reward $\mu_a$ as

$$\widetilde{\mu}_{a,t} := \frac{1}{\sum_{s=1}^{t-1} \mathbb{1}[A_s = a]} \sum_{s=1}^{t-1} \mathbb{1}[A_s = a] Y_{a,s}.$$

Also, we define a second moment estimator $\widetilde{\zeta}_{a,t} := (\sum_{s=1}^{t-1} \mathbb{1}[A_s = a])^{-1} \sum_{s=1}^{t-1} \mathbb{1}[A_s = a] Y_{a,s}^2$, and a root of variance estimator $\widetilde{\sigma}_{a,t} = \{\widetilde{\zeta}_{a,t} - (\widetilde{\mu}_{a,t})^2\}^{1/2}$. Then, we define the estimator $\widehat{\sigma}_{a,t} = \mathrm{thre}(\widetilde{\sigma}_{a,t}; \underline{C}_{\sigma^2}^{1/2}, 1/\overline{C}_{\sigma^2})$ with the constant $C_{\sigma^2} > 0$, defined in Section 1. Note that this truncation is introduced for a technical purpose to draw each arm infinitely many times as $T \to \infty$ and avoid the estimators of $\mu_1^*$ and $\mu_2^*$, defined below, diverging to infinity. We just use a sufficiently small value for $C_{\sigma^2} > 0$. Also, we define the estimator $\widehat{w}_{1,t}$ and $\widehat{w}_{2,t}$ as

$$\widehat{w}_{1,t} := \frac{\widehat{\sigma}_{1,t}}{\widehat{\sigma}_{1,t} + \widehat{\sigma}_{2,t}}, \quad \text{and} \quad \widehat{w}_{2,t} := 1 - \widehat{w}_{1,t}. \tag{1}$$

In each round $t \geq 3$, we draw arm $A_t = 1$ with probability $\widehat{w}_{1,t}$ and $A_t = 2$ with probability $\widehat{w}_{2,t}$.

We note the possibility of increasing the number of initialization rounds, although our strategy utilizes only the first two rounds for this purpose. The additional rounds of initialization serve to stabilize the sampling rule in practical applications, akin to the concept of the forced-sampling (Garivier & Kaufmann, 2016). We can change the number of initialization rounds if the condition $\widehat{w}_{a,t} \xrightarrow{\text{a.s.}} w_a^*$ is satisfied as $t \to \infty$ for every $a \in \{1, 2\}$, which is crucial for our theoretical analysis. For instance, instead of using $\widehat{w}_{1,t}$ directly, an alternative arm-drawing probability could be defined as $\widetilde{w}_{1,t} = \alpha_t/2 + (1 - \alpha_t)\widehat{w}_{1,t}$, assuming $\alpha_t \in [0, 1]$ and converges to zero as $t$ approaches infinity (here, we define $\widetilde{w}_{2,t} = 1 - \widetilde{w}_{1,t}$). Moreover, the number of initialization rounds can be made dependent upon the number of arms without impacting the theoretical outcomes.

---

[1]Similar strategies are often used in the context of the average treatment effect estimation by an adaptive experiment (van der Laan, 2008; Kato et al., 2020).

---

**Algorithm 1** NA-AIPW Strategy

---

**Parameter:** Positive constants $C_\mu$ and $C_{\sigma^2}$.
**Initialization:**
At $t = 1$, sample $A_t = 1$; at $t = 2$, sample $A_t = 2$. For $a \in \{1, 2\}$, set $\widehat{w}_{a,1} = \widehat{w}_{a,2} = 0.5$.
**for** $t = 3$ to $T$ **do**
    Construct $\widehat{w}_{a,t}$ following equation 1.
    Draw $A_t = 1$ with probability $\widehat{w}_{1,t}$ and $A_t = 2$ with probability $\widehat{w}_{2,t} = 1 - \widehat{w}_{1,t}$.
    Observe $Y_t$.
**end for**
Construct $\widehat{\mu}_{a,T}^{\mathrm{AIPW}}$ for $a \in \{1, 2\}$. following equation 2.
Recommend $\widehat{a}_T$ following equation 3.

---

### 3.3 Recommendation Rule with the Augmented Inverse Probability Weighting (AIPW) Estimator

We present our recommendation rule. In the recommendation phase after round $T$, we estimate $\mu_a^*$ for each $a \in \{1, 2\}$ and recommend an arm with the bigger estimated expected reward. With a truncated version of the estimated expected reward $\widehat{\mu}_{a,t} := \mathrm{thre}(\widetilde{\mu}_{a,t}, -C_\mu, C_\mu)$ with some predetermined constant $C_\mu > 0$, defined in Section 1, we define the *augmented inverse probability weighting* (AIPW) estimator of $\mu_a^*$ for each $a \in \{1, 2\}$ as

$$\widehat{\mu}_{a,T}^{\mathrm{AIPW}} := \frac{1}{T} \sum_{t=1}^{T} \psi_{a,t}, \qquad \text{where } \psi_{a,t} := \frac{\mathbb{1}[A_t = a]\big(Y_{a,t} - \widehat{\mu}_{a,t}\big)}{\widehat{w}_{a,t}} + \widehat{\mu}_{a,t}. \tag{2}$$

At the end of the experiment (after the round $t = T$), we recommend $\widehat{a}_T$ as

$$\widehat{a}_T := \begin{cases} 1 & \text{if} \quad \widehat{\mu}_{1,T}^{\mathrm{AIPW}} \geq \widehat{\mu}_{2,T}^{\mathrm{AIPW}}, \\ 2 & \text{otherwise.} \end{cases} \tag{3}$$

We adopt the AIPW estimator for our strategy because it has several advantages. First, the AIPW estimator has the property of semiparametric efficiency, which indicates that it has the smallest asymptotic variance among a certain class (Hahn, 1998). The property is necessary to prove that the strategy using the AIPW estimator is optimal, which means the misidentification probability is small enough to achieve its lower bound. The second reason is more technical; the AIPW estimator simplifies the theoretical analysis (see Section 6.3). Specifically, we can decompose an error by the AIPW estimator into a sum of random variables with martingale properties, making it suitable for analysis using the central limit theorem. This property is unique to the AIPW estimator but not to naive estimators such as an empirical average. Details will be given in Section 5.

We provide the pseudo-code for our proposed strategy in Algorithm 1. Note that we introduce $C_\mu$ and $C_{\sigma^2}$ for technical purposes to bound the estimators and any large positive value can be used.

## 4 Misidentification Probability and Asymptotic Optimality

In this section, we show the following upper bound of the misspecification probability of the NA-AIPW strategy, which also implies that the strategy is asymptotically optimal.

**Theorem 4.1** (Upper bound of the NA-AIPW strategy). *For any $P^* \in \mathcal{P}^{\mathrm{G}}$ with known constants $C_\mu, C_{\sigma^2} > 0$ independent of $T$, if $\{(Y_{1,t}, Y_{2,t})\}_{t \in [T]}$ is generated from $P^*$, the following holds as $\Delta \to 0$:*

$$\liminf_{T \to \infty} -\frac{1}{T} \log \mathbb{P}_{P^*}\big(\widehat{a}_T^{\mathrm{AIPW}} \neq a^\star(P^*)\big) \geq \frac{\Delta^2}{2(\sigma_1 + \sigma_2)^2} - o\big(\Delta^2\big).$$

We note again that $C_\mu$ and $C_{\sigma^2}$ are introduced for technical purpose. The constant $C_\mu$ is introduced to guarantee the boundedness of the estimators, and it is sufficient to use a sufficiently large value for it. The

constant $C_{\sigma^2}$ is used to draw each arm infinitely many times as $T \to \infty$ and avoid the estimators of the means $\mu^* 1$ and $\mu_2^*$ diverging to infinity, and it is sufficient to use a sufficiently small value for $C_{\sigma^2} > 0$.

Note that the lower bound of $-\frac{1}{T} \log \mathbb{P}_{P^*} \left( \widehat{a}_T^{\mathrm{AIPW}} \neq a^\star(P^*) \right)$ implies the upper bound of $\mathbb{P}_{P^*} \left( \widehat{a}_T^{\mathrm{AIPW}} \neq a^\star(P^*) \right)$. This theorem implies us to evaluate the probability of misidentification up to the constant term, even when it is exponentially small, as $\Delta \to 0$.

This result directly implies the asymptotic optimality of the NA-AIPW strategy. As $\Delta \to 0$, the upper bound matches the lower bound in Proposition 2.1. This asymptotic optimality result suggests that the estimation error of the target allocation ratio (variances) $w^*$ is negligible when $\Delta \to 0$. This is because the estimation error is insignificant compared to the challenges of identifying the best arm due to the small gap.

Although studies, such as Ariu et al. (2021), Qin (2022), and Degenne (2023), point out the non-existence of the optimal strategies in fixed-budget BAI against the lower bound shown by Kaufmann et al. (2016), our result does not yield a contradiction. Existing impossibility results discuss the existence of a strategy that violates the lower bound. Note that the lower bounds in Kaufmann et al. (2016) are applicable to any instances in the bandit models (with some regularity conditions). In other words, if we consider the lower bound in Kaufmann et al. (2016) for all instances, there exists an instance under which there exists a strategy whose lower bound is larger than the lower bound derived by Kaufmann et al. (2016). In contrast, we only consider bandit models where $\Delta \to 0$. Our result implies that if we restrict bandit models, the upper bounds of our strategy within the restricted bandit models match the lower bound. Because our optimality is limited to a case where $\Delta \to 0$, we refer to our optimality as asymptotic optimality under the small-gap regime or local asymptotic optimality.

We conjecture that even if we replace the AIPW estimator with the sample average estimator, defined as $\widetilde{\mu}_{a,t} = (\sum_{s=1}^{t-1} \mathbb{1}[A_s = a])^{-1} \sum_{s=1}^{t-1} \mathbb{1}[A_s = a] Y_{a,s}$ in Section 3.2, the upper bound of the strategy still matches the lower bound. However, the proof is an open issue. Hirano et al. (2003) and Hahn et al. (2011) show that the sample average estimator $\widetilde{\mu}_{a,t}$ and the AIPW estimator have the same asymptotic variance (or asymptotic distribution). To show the result, we need to employ empirical process arguments. One of the problems in extending the result to analysis for BAI is that their result focuses on the asymptotic distribution, not the tail probability. Therefore, to show the asymptotic optimality of the strategy with the sample average in the sense of the probability of misidentification, we need to modify the result in Hirano et al. (2003) and Hahn et al. (2011) to analyze the tail probability.

## 5 Proof of Theorem 4.1

To show Theorem 4.1, we derive the upper bound of $\mathbb{P}_{P^*} \left( \widehat{\mu}_{a^\star(P^*),T}^{\mathrm{AIPW}} \leq \widehat{\mu}_{b,T}^{\mathrm{AIPW}} \right)$ for $b \in \{1,2\} \backslash \{a^\star(P^*)\}$, which is equivalent to $\mathbb{P}_{P^*} \left( \widehat{a}_T^{\mathrm{AIPW}} \neq a^\star(P^*) \right)$. Without loss of generality, we assume that $a^\star(P^*) = 1$ and $b = 2$. Let us define $V := \frac{\sigma_1^2}{w_1^*} + \frac{\sigma_2^2}{w_2^*} = (\sigma_1 + \sigma_2)^2$ and

$$\Psi_t := \frac{\psi_{1,t} - \psi_{2,t} - \Delta}{\sqrt{V}}.$$

Therefore, in the following parts, we aim to derive the upper bound of $\mathbb{P}_{P^*} \left( \widehat{\mu}_{a^\star(P^*),T}^{\mathrm{AIPW}} \leq \widehat{\mu}_{b,T}^{\mathrm{AIPW}} \right) = \mathbb{P}_{P^*} \left( \widehat{\mu}_{1,T}^{\mathrm{AIPW}} \leq \widehat{\mu}_{2,T}^{\mathrm{AIPW}} \right) = \mathbb{P}_{P^*} \left( \sum_{t=1}^T \Psi_t \leq -\frac{T\Delta}{\sqrt{V}} \right)$. Let $\mathbb{E}_P$ be the expectation under $P \in \mathcal{P}^{\mathrm{G}}$. We derive the upper bound using the Chernoff bound. This proof is partially inspired by techniques in Hadad et al. (2021), and Kato et al. (2020).

First, because there exists a constant $C > 0$ independent of $T$ such that $\widehat{w}_{a,t} > C$ by construction, the following lemma holds.

**Lemma 5.1.** *For any $P^* \in \mathcal{P}^{\mathrm{G}}$ and all $a \in \{1,2\}$, $\widehat{\mu}_{a,t} \xrightarrow{\mathrm{a.s}} \mu_a^*$ and $\widehat{\sigma}_a^2 \xrightarrow{\mathrm{a.s}} \sigma_a^2$.*

Furthermore, from $\widehat{\sigma}_a^2 \xrightarrow{\mathrm{a.s}} \sigma_a^2$ and continuous mapping theorem, for any $P^* \in \mathcal{P}^{\mathrm{G}}$ and all $a \in \{1,2\}$, $\widehat{w}_{a,t} \xrightarrow{\mathrm{a.s}} w_{a,t}^*$.

**Step 1: Sequence $\{\Psi_t\}_{t=1}^T$ is a martingale difference sequence (MDS)**

We prove that $\{\Psi_t\}_{t=1}^T$ is an MDS; that is, $\mathbb{E}_{P^*}[\Psi_t|\mathcal{F}_{t-1}] = 0$. Although this fact is well-known in the literature of causal inference (van der Laan, 2008; Hadad et al., 2021; Kato et al., 2020), we show the proof for the sake of completeness.

**Lemma 5.2.** *For any $P^* \in \mathcal{P}^{\mathrm{G}}$, $\{\Psi_t\}_{t=1}^T$ is an MDS.*

*Proof.* For each $t \in [T]$, it holds that

$$
\sqrt{V}\,\mathbb{E}_{P^*}[\Psi_t|\mathcal{F}_{t-1}]
$$
$$
= \mathbb{E}_{P^*}\left[\frac{\mathbb{1}[A_t = 1](Y_{1,t} - \widehat{\mu}_{1,t})}{\widehat{w}_{1,t}} + \widehat{\mu}_{1,t}\Big|\mathcal{F}_{t-1}\right] - \mathbb{E}_{P^*}\left[\frac{\mathbb{1}[A_t = 2](Y_{2,t} - \widehat{\mu}_{2,t})}{\widehat{w}_{2,t}} + \widehat{\mu}_{2,t}\Big|\mathcal{F}_{t-1}\right] - \Delta
$$
$$
= \frac{\widehat{w}_{1,t}(\mu_1^* - \widehat{\mu}_{1,t})}{\widehat{w}_{1,t}} + \widehat{\mu}_{1,t} - \frac{\widehat{w}_{2,t}(\mu_2^* - \widehat{\mu}_{2,t})}{\widehat{w}_{2,t}} + \widehat{\mu}_{2,t} - \Delta = \{(\mu_1^* - \mu_2^*) - (\mu_1^* - \mu_2^*)\} = 0
$$

$\square$

**Step 2: Evaluation by using the Chernoff Bound with Martingales**

By applying the Chernoff bound, for any $v < 0$ and any $\lambda < 0$, it holds that

$$
\mathbb{P}_{P^*}\left(\frac{1}{T}\sum_{t=1}^T \Psi_t \leq v\right) \leq \mathbb{E}_{P^*}\left[\exp\left(\lambda \sum_{t=1}^T \Psi_t\right)\right]\exp(-T\lambda v).
$$

From the Chernoff bound and a property of an MDS, we have

$$
\mathbb{E}_{P^*}\left[\exp\left(\lambda \sum_{t=1}^T \Psi_t\right)\right] = \mathbb{E}_{P^*}\left[\prod_{t=1}^T \mathbb{E}_{P^*}\left[\exp(\lambda\Psi_t)|\mathcal{F}_{t-1}\right]\right] = \mathbb{E}_{P^*}\left[\exp\left(\sum_{t=1}^T \log \mathbb{E}_{P^*}\left[\exp(\lambda\Psi_t)|\mathcal{F}_{t-1}\right]\right)\right].
$$

Then, the Taylor expansion around $\lambda = 0$ yields

$$
\log \mathbb{E}_{P^*}\left[\exp(\lambda\Psi_t)|\mathcal{F}_{t-1}\right] = \frac{\lambda^2}{2}\mathbb{E}_{P^*}\left[\Psi_t^2|\mathcal{F}_{t-1}\right] + o(\lambda^2) \tag{4}
$$

as $\lambda \to 0$. This is given as follows. Since $\mathbb{E}_{P^*}\left[\Psi_t^k \exp(\lambda\Psi_t)|\mathcal{F}_{t-1}\right]$ are finite everywhere in an open interval $(0, \infty)$ for $k = 1, 2, 3$, the Taylor expansion yields the following (for the details, see the textbook such as page 75 in Bulmer (1967) and Theorem 5.19 in Apostol (1974)):

$$
\mathbb{E}_{P^*}\left[\exp(\lambda\Psi_t)|\mathcal{F}_{t-1}\right] = 1 + \sum_{k=1}^2 \mathbb{E}_{P^*}\left[\Psi_t^k/k!|\mathcal{F}_{t-1}\right] + o(\lambda^2)
$$

as $\lambda \to 0$. Note that the finiteness of $\mathbb{E}_{P^*}\left[\Psi_t^k \exp(\lambda\Psi_t)|\mathcal{F}_{t-1}\right]$ comes from the following in $\Psi_t$: (i) $Y_{a,t}$ is a Gaussian random variable, (ii) $\widehat{\mu}_{a,t}$ and $\widehat{w}_{a,t}$ are bounded random variables by our truncation, and (iii) the lower bound of $\widehat{w}$ is given by $C_{\sigma^2}$. By using the Taylor expansion again, we approximate $\log(1 + z)$ around $z = 0$ as $\log(1 + z) = z - z^2/2 + z^3/3 - \cdots$. Therefore, we have

$$
\log \mathbb{E}_{P^*}\left[\exp(\lambda\Psi_t)|\mathcal{F}_{t-1}\right]
$$
$$
= \left\{\lambda\mathbb{E}_{P^*}[\Psi_t|\mathcal{F}_{t-1}] + \frac{\lambda^2}{T}\mathbb{E}_{P^*}\left[\Psi_t^2/2!|\mathcal{F}_{t-1}\right] + o(\lambda^2)\right\} - \frac{1}{2}\left\{\lambda\mathbb{E}_{P^*}[\Psi_t|\mathcal{F}_{t-1}] + o(\lambda)\right\}^2
$$
$$
= \frac{\lambda^2}{T}\mathbb{E}_{P^*}\left[\Psi_t^2/2!|\mathcal{F}_{t-1}\right] + o(\lambda^2)
$$

as $\lambda \to 0$. Here, we used $\mathbb{E}_{P^*}[\Psi_t|\mathcal{F}_{t-1}] = 0$. Thus, the equation 4 holds.

**Step 3: Convergence of the Second Moment**

We next show $\mathbb{E}_{P^*}\left[\Psi_t^2|\mathcal{F}_{t-1}\right] - 1 \xrightarrow{\text{a.s}} 0$. This result is a direct consequence of Lemma 5.1.

**Lemma 5.3.** *For any $P^* \in \mathcal{P}^{\mathrm{G}}$, we have*

$$\mathbb{E}_{P^*}\left[\Psi_t^2|\mathcal{F}_{t-1}\right] - 1 \xrightarrow{\text{a.s}} 0 \qquad \text{as} \ \ t \to \infty.$$

*Proof.* We have

$$
V\mathbb{E}_{P^*}\left[\Psi_t^2|\mathcal{F}_{t-1}\right] = \mathbb{E}_{P^*}\left[(\psi_{1,t} - \psi_{2,t} - \Delta)^2 \Big|\mathcal{F}_{t-1}\right]
$$

$$
= \mathbb{E}_{P^*}\left[\left(\frac{\mathbb{1}[A_t = 1]\left(Y_{1,t} - \widehat{\mu}_{1,t}\right)}{\widehat{w}_{1,t}} - \frac{\mathbb{1}[A_t = 2]\left(Y_{2,t} - \widehat{\mu}_{2,t}\right)}{\widehat{w}_{2,t}}\right)^2\right.
$$

$$
+ 2\left(\frac{\mathbb{1}[A_t = a^\star(P^*)]\left(Y_{1,t} - \widehat{\mu}_{1,t}\right)}{\widehat{w}_{1,t}} - \frac{\mathbb{1}[A_t = a]\left(Y_{2,t} - \widehat{\mu}_{2,t}\right)}{\widehat{w}_{2,t}}\right) \times (\widehat{\mu}_{1,t} - \widehat{\mu}_{2,t} - (\mu_1^* - \mu_2^*))
$$

$$
\left. + (\widehat{\mu}_{1,t} - \widehat{\mu}_{2,t} - (\mu_1^* - \mu_2^*))^2 \,|\mathcal{F}_{t-1}\right]
$$

$$
= \mathbb{E}_{P^*}\left[\frac{\mathbb{1}[A_t = 1]\left(Y_{1,t} - \widehat{\mu}_{1,t}\right)^2}{\left(\widehat{w}_{1,t}\right)^2} + \frac{\mathbb{1}[A_t = 2]\left(Y_{2,t} - \widehat{\mu}_{2,t}\right)^2}{\left(\widehat{w}_{2,t}\right)^2}\right.
$$

$$
+ 2\left(\frac{\mathbb{1}[A_t = 1]\left(Y_{1,t} - \widehat{\mu}_{1,t}\right)}{\widehat{w}_{1,t}} - \frac{\mathbb{1}[A_t = a]\left(Y_{2,t} - \widehat{\mu}_{2,t}\right)}{\widehat{w}_{2,t}}\right)(\widehat{\mu}_{1,t} - \widehat{\mu}_{2,t} - (\mu_1^* - \mu_2^*))
$$

$$
\left. + \left((\widehat{\mu}_{1,t} - \widehat{\mu}_{2,t}) - (\mu_1^* - \mu_2^*)\right)^2 \,|\mathcal{F}_{t-1}\right]
$$

$$
= \sum_{a \in \{1,2\}} \mathbb{E}_{P^*}\left[\frac{\left(Y_{a,t} - \widehat{\mu}_{a,t}\right)^2}{\left(\widehat{w}_{a,t}\right)^2}|\mathcal{F}_{t-1}\right] - \mathbb{E}_{P^*}\left[\left((\widehat{\mu}_{1,t} - \widehat{\mu}_{2,t}) - \left(\mu_1^* - \mu_2^*\right)\right)^2 |\mathcal{F}_{t-1}\right].
$$

Here, for $a \in \{1, 2\}$, the followings hold:

$$
\mathbb{E}_{P^*}\left[\frac{\mathbb{1}[A_t = a]\left(Y_{a,t} - \widehat{\mu}_{a,t}\right)^2}{\left(\widehat{w}_{a,t}\right)^2}|\mathcal{F}_{t-1}\right] = \mathbb{E}_{P^*}\left[\frac{\left(Y_{a,t} - \widehat{\mu}_{a,t}\right)^2}{\widehat{w}_{a,t}}|\mathcal{F}_{t-1}\right] = \frac{\mathbb{E}_{P^*}[(Y_{a,t})^2] - 2\mu_a^*\widehat{\mu}_{a,t} + (\widehat{\mu}_{a,t})^2}{\widehat{w}_{a,t}}
$$

$$
= \frac{\mathbb{E}_{P^*}[(Y_{a,t})^2] - (\mu_a^*)^2 + (\mu_a^* - \widehat{\mu}_{a,t})^2}{\widehat{w}_{a,t}} = \frac{\sigma_a^2 + (\mu_a^* - \widehat{\mu}_{a,t})^2}{\widehat{w}_{a,t}},
$$

and

$$
\mathbb{E}_{P^*}\left[\frac{\mathbb{1}[A_t = a]\left(Y_{a,t} - \widehat{\mu}_{2,t}\right)^2}{\left(\widehat{w}_{1,t}\right)^2}\frac{\mathbb{1}[A_t = a]\left(Y_{a,t} - \widehat{\mu}_{2,t}\right)^2}{\left(\widehat{w}_{2,t}\right)^2}|\mathcal{F}_{t-1}\right] = 0,
$$

where we used $\mathbb{E}_{P^*}[(Y_{a,t})^2|x] - (\mu_a^*)^2 = \sigma_a^2$. Therefore, the following holds:

$$
\mathbb{E}_{P^*}\left[\frac{\left(Y_{1,t} - \widehat{\mu}_{1,t}\right)^2}{\widehat{w}_{1,t}}|\mathcal{F}_{t-1}\right] + \mathbb{E}_{P^*}\left[\frac{\left(Y_{2,t} - \widehat{\mu}_{2,t}\right)^2}{\widehat{w}_{2,t}}|\mathcal{F}_{t-1}\right] - \mathbb{E}_{P^*}\left[\left((\widehat{\mu}_{1,t} - \widehat{\mu}_{2,t}) - (\mu_1^* - \mu_2^*)\right)^2 |\mathcal{F}_{t-1}\right]
$$

$$
= \mathbb{E}_{P^*}\left[\frac{\sigma_1^2 + (\mu_1^* - \widehat{\mu}_{1,t})^2}{\widehat{w}_{1,t}}\right] + \mathbb{E}_{P^*}\left[\frac{\sigma_2^2 + (\mu_2^* - \widehat{\mu}_{2,t})^2}{\widehat{w}_{2,t}}\right] - \mathbb{E}_{P^*}\left[\left((\widehat{\mu}_{1,t} - \widehat{\mu}_{2,t}) - (\mu_1^* - \mu_2^*)\right)^2\right].
$$

Because $\widehat{\mu}_{a,t} \xrightarrow{\text{a.s.}} \mu_a^*$ and $\widehat{w}_{a,t} \xrightarrow{\text{a.s.}} w_a^*$, we have

$$
\lim_{t \to \infty}\left|\left(\frac{\sigma_1^2 + (\mu_1^* - \widehat{\mu}_{1,t})^2}{\widehat{w}_{1,t}}\right) + \left(\frac{\sigma_2^2 + (\mu_2^* - \widehat{\mu}_{2,t})^2}{\widehat{w}_{2,t}}\right) - \left((\widehat{\mu}_{1,t} - \widehat{\mu}_{2,t}) - (\mu_1^* - \mu_2^*)\right)^2\right.
$$

$$
-\left(\frac{\sigma_1^2}{w_1^*} + \frac{\sigma_a^2}{w_2^*} + \left((\mu_1^* - \mu_2^*) - (\mu_1^* - \mu_2^*)\right)^2\right)\Bigg|
$$

$$
\leq \lim_{t\to\infty}\left|\frac{\sigma_1^2}{\widehat{w}_{1,t}} - \frac{\sigma_1^2}{w_1^*}\right| + \lim_{t\to\infty}\left|\frac{\sigma_2^2}{\widehat{w}_{2,t}} - \frac{\sigma_2^2}{w_2^*}\right| + \lim_{t\to\infty}\frac{(\mu_1^* - \widehat{\mu}_{1,t})^2}{\widehat{w}_{1,t}} + \lim_{t\to\infty}\frac{(\mu_2^* - \widehat{\mu}_{2,t})^2}{\widehat{w}_{2,t}}
$$

$$
+ \lim_{t\to\infty}\left|\left((\widehat{\mu}_{1,t} - \widehat{\mu}_{2,t}) - (\mu_1^* - \mu_2^*)\right)^2 - \left((\mu_1^* - \mu_2^*) - (\mu_1^* - \mu_2^*)\right)^2\right| = 0,
$$

with probability 1. Therefore, from Lebesgue's dominated convergence theorem, we obtain

$$
V\mathbb{E}_{P^*}\left[\Psi_t^2|\mathcal{F}_{t-1}\right] - V
$$

$$
= \mathbb{E}_{P^*}\left[\frac{\sigma_1^2 + (\mu_1^* - \widehat{\mu}_{1,t})^2}{\widehat{w}_{1,t}}|\mathcal{F}_{t-1}\right] + \mathbb{E}_{P^*}\left[\frac{\sigma_a^2 + (\mu_2^* - \widehat{\mu}_{2,t})^2}{\widehat{w}_{2,t}}|\mathcal{F}_{t-1}\right]
$$

$$
- \mathbb{E}_{P^*}\left[\left((\widehat{\mu}_{1,t} - \widehat{\mu}_{2,t}) - (\mu_1^* - \mu_2^*)\right)^2|\mathcal{F}_{t-1}\right] - \mathbb{E}_{P^*}\left[\frac{\sigma_1^2}{w_1^*} + \frac{\sigma_2^2}{w_2^*} + \left((\mu_1^* - \mu_2^*) - (\mu_1^* - \mu_2^*)\right)^2|\mathcal{F}_{t-1}\right]
$$

$$
\xrightarrow{\text{a.s.}} 0.
$$

$\square$

This lemma immediately yields the following lemma.

**Lemma 5.4.** *For any $P^* \in \mathcal{P}^{\mathrm{G}}$ and any $\epsilon > 0$, there exists $t(\epsilon) > 0$ such that for all $T > t(\epsilon)$, we have*

$$
\frac{1}{T}\sum_{t=1}^{T}\left|\mathbb{E}_{P^*}\left[\Psi_t^2|\mathcal{F}_{t-1}\right] - 1\right| < \epsilon
$$

*with probability one.*

This result is a variant of the Cesàro lemma for a case with almost sure convergence. For completeness, we show the proof, which is based on the proof of Lemma 10 in Hadad et al. (2021).

*Proof.* Let $u_t$ be $u_t = \mathbb{E}_{P^*}\left[\Psi_t^2|\mathcal{F}_{t-1}\right] - 1$. Note that $\frac{1}{T}\sum_{t=1}^{T}\mathbb{E}_{P^*}\left[\Psi_t^2|\mathcal{F}_{t-1}\right] - 1 = \frac{1}{T}\sum_{t=1}^{T}u_t$.

From the proof of Lemma 5.3, we can find that $u_t$ is a bounded random variable. Recall that

$$
V\mathbb{E}_{P^*}\left[\Psi_t^2|\mathcal{F}_{t-1}\right]
$$

$$
= \mathbb{E}_{P^*}\left[\frac{\sigma_1^2 + (\mu_1^* - \widehat{\mu}_{1,t})^2}{\widehat{w}_{1,t}}|\mathcal{F}_{t-1}\right] + \mathbb{E}_{P^*}\left[\frac{\sigma_2^2 + (\mu_2^* - \widehat{\mu}_{2,t})^2}{\widehat{w}_{2,t}}|\mathcal{F}_{t-1}\right] - \mathbb{E}_{P^*}\left[\left((\widehat{\mu}_{1,t} - \widehat{\mu}_{2,t}) - (\mu_1^* - \mu_2^*)\right)^2|\mathcal{F}_{t-1}\right].
$$

We assumed that $(\mu_1^*, \mu_2^*, \widehat{\mu}_{1,t}, \widehat{\mu}_{2,t}, \widehat{w}_{1,t}, \widehat{w}_{2,t})$ are all bounded random variables. Let $C$ be a constant independent of $T$ such that $|u_t| < C$ for all $t \in \mathbb{N}$.

Almost-sure convergence of $u_t$ to zero as $t \to \infty$ implies that for any $\epsilon' > 0$, there exists $t(\epsilon)$ such that $|u_t| < \epsilon'$ for all $t \geq t(\epsilon')$ with probability one. Let $\mathcal{E}(\epsilon')$ denote the event in which this happens; that is, $\mathcal{E}(\epsilon') = \{|u_t| < \epsilon' \quad \forall\, t \geq t(\epsilon')\}$. Under this event, for $T > t(\epsilon')$, the following holds:

$$
\frac{1}{T}\sum_{t=1}^{T}|u_t| \leq \frac{1}{T}\sum_{t=1}^{t(\epsilon')}C + \frac{1}{T}\sum_{t=t(\epsilon')+1}^{T}\epsilon = \frac{1}{T}t(\epsilon')C + \epsilon',
$$

where $\frac{1}{T}t(\epsilon')C \to 0$ as $T \to \infty$.

Therefore, for any $\epsilon > 0$, there exists $t(\epsilon)$ such that for all $T > t(\epsilon)$, $\frac{1}{T}\sum_{t=1}^{T}|u_t| < \epsilon$ holds with probability one. $\square$

**Step 4: Tail Bound with the Approximated Second Moment**

Let $v = \lambda$. Then, we have

$$\mathbb{P}_{P^*}\left(\sum_{t=1}^T \Psi_t \leq Tv\right) \leq \mathbb{E}_{P^*}\left[\exp\left(-\frac{T\lambda^2}{2} + \frac{\lambda^2}{2}\left\{\sum_{t=1}^T \mathbb{E}_{P^*}\left[\Psi_t^2|\mathcal{F}_{t-1}\right] - 1\right\} + To\left(\lambda^2\right)\right)\right].$$

From Lemma 5.4, for any $\epsilon > 0$, there exists $t(\epsilon) > 0$ such that for all $T > t(\epsilon)$, we have

$$\mathbb{E}_{P^*}\left[\exp\left(-\frac{T\lambda^2}{2} + \frac{\lambda^2}{2}\left\{\sum_{t=1}^T \mathbb{E}_{P^*}\left[\Psi_t^2|\mathcal{F}_{t-1}\right] - 1\right\} + To\left(\lambda^2\right)\right)\right]$$

$$= \exp\left(-\frac{T\lambda^2}{2} + To\left(\lambda^2\right)\right) \mathbb{E}_{P^*}\left[\exp\left(\frac{\lambda^2}{2}\left\{\sum_{t=1}^T \mathbb{E}_{P^*}\left[\Psi_t^2|\mathcal{F}_{t-1}\right] - 1\right\}\right)\right]$$

$$\leq \exp\left(-\frac{T\lambda^2}{2} + To\left(\lambda^2\right)\right) \exp\left(\frac{\lambda^2}{2}T\epsilon\right) = \exp\left(-\frac{T\lambda^2}{2} + To\left(\lambda^2\right) + \frac{\lambda^2}{2}T\epsilon\right).$$

**Step 5: Final Step of the Proof of Theorem 4.1**

For any $\epsilon > 0$, there exists $t(\epsilon) > 0$ such that for all $T > t(\epsilon)$, we obtainz

$$-\frac{1}{T}\log \mathbb{P}_{P^*}\left(\widehat{\mu}_{1,T}^{\mathrm{AIPW}} \leq \widehat{\mu}_{2,T}^{\mathrm{AIPW}}\right) \geq -\left\{-\frac{\lambda^2}{2} + o\left(\lambda^2\right) + \frac{\lambda^2}{2}\epsilon\right\} = \frac{\lambda^2}{2} - o\left(\lambda^2\right) - \frac{\lambda^2}{2}\epsilon,$$

as $\lambda \to 0$.

Let $\lambda = -\frac{\Delta}{\sqrt{V}}$. Then, we have

$$-\frac{1}{T}\log \mathbb{P}_{P^*}\left(\widehat{\mu}_T^{\mathrm{AIPW},a^*(P)} \leq \widehat{\mu}_T^{\mathrm{AIPW},a}\right) \geq \frac{\Delta^2}{2V} - o\left(\frac{\Delta^2}{V}\right) - \frac{\epsilon\Delta^2}{2V},$$

as $\Delta \to 0$. By letting $\Delta \to 0$ and $T \to \infty$, and then letting $\epsilon \to 0$ independently of $T$ and $\Delta$, we have

$$-\frac{1}{T}\log \mathbb{P}_{P^*}\left(\widehat{\mu}_{1,T}^{\mathrm{AIPW}} \leq \widehat{\mu}_{2,T}^{\mathrm{AIPW}}\right) \geq \frac{\Delta^2}{2V} - o\left(\Delta^2\right).$$

Thus, the proof is complete.

# 6 Discussion

In this section, we discuss related topics.

## 6.1 Neyman Allocation with Unknown Variances

For two-armed Gaussian bandits with known variances, Chen et al. (2000), Glynn & Juneja (2004), and Kaufmann et al. (2016) conclude that sampling each arm with a proportion of the standard deviation is optimal, which corresponds to the Neyman allocation Neyman (1934).

The Neyman allocation with unknown variances has been long studied in various fields. van der Laan (2008) and Hahn et al. (2011) develop algorithms for estimating the gap parameter $\Delta$ itself in an adaptive experiment with the Neyman allocation. They estimate the variances and show their algorithms' optimalities under the framework of semiparametric efficiency, which closely connects to the Gaussian approximation of estimators using the central limit theorem. Although they show their optimality under the framework, they do not investigate the asymptotic optimality in the large-deviation framework. Tabord-Meehan (2022), Kato et al. (2020), and Zhao (2023) also attempt to adrress related problems.

Jourdan et al. (2023) examines BAI with unknown variances in a fixed-confidence setting. Beyond the difference in settings (we focus on fixed-budget BAI), the methods of deriving lower bounds differ between our approach and theirs. They determine the lower bound while incorporating the assumption that the variances are unknown. Moreover, under a large-gap regime ($\Delta$ is fixed), they confirm a discrepancy between the lower bounds when variances are known versus unknown. Specifically, they consider alternative hypotheses related to both variances and means. In contrast, the lower bounds presented by Kaufmann et al. (2016) and ourselves are based on alternative hypotheses with fixed variances. While Jourdan et al. (2023) suggests that the upper bounds of strategies with unknown variances cannot align with the lower bound when variances are known, our findings indicate a match under the small-gap regime.

## 6.2 Necessity of the Small-Gap Regime

First, we discuss the necessity of the small-gap regime.

**Estimation error of the variances.** The most critical reason we employ the small-gap regime is that the estimation error of the variances cannot be ignored in evaluating the probability of misidentification. To clarify this point, we review the probability of misidentification when we know the variances.

**Probability of misidentification of the Kaufmann et al. (2016)'s strategy with known variances.** Kaufmann et al. (2016) proposes drawing arm $a$ in $\frac{\sigma^a}{\sigma^1+\sigma^2}T = w_a^*T$ rounds (for simplicity, we deal with $w_a^*T$ as an integer). Without loss of generality, we consider draw arm 1 in the first $w_1^*T$ rounds and draw arm 2 in the following $T - w_1^*T = w_2^*T$ rounds. Then, they estimate the best arm as $\widehat{a}_T^{\mathrm{KCG}} := \arg\max_{a \in \{1,2\}} \widehat{\mu}_{a,T}^{\mathrm{SA}}$, where $\widehat{\mu}_{a,T}^{\mathrm{SA}}$ is the sample average defined as

$$\widehat{\mu}_{a,T}^{\mathrm{SA}} := \frac{1}{\sum_{t=1}^{T} \mathbb{1}[A_t^{\mathrm{KCG}} = a]} \sum_{t=1}^{T} \mathbb{1}[A_t^{\mathrm{KCG}} = a]Y_t, \tag{5}$$

where $A_t^{\mathrm{KCG}}$ denotes an arm drawn by the Kaufmann et al. (2016)'s strategy. For the strategy, they show that its probability of misidentification is given as

$$\liminf_{T \to \infty} -\frac{1}{T} \log \mathbb{P}_{P^*} \left( \widehat{a}_T^{\mathrm{KCG}} \neq a^\star(P^*) \right) \geq \frac{\Delta^2}{2(\sigma_1 + \sigma_2)^2}. \tag{6}$$

Note that this upper bound comes from the upper bound of

$$\liminf_{T \to \infty} -\frac{1}{T} \log \mathbb{P}_{P^*} \left( \widehat{a}_T^{\mathrm{KCG}} \neq a^\star(P^*) \right) = \liminf_{T \to \infty} -\frac{1}{T} \log \mathbb{P}_{P^*} \left( \widehat{\mu}_{1,T}^{\mathrm{SA}} - \widehat{\mu}_{2,T}^{\mathrm{SA}} \leq 0 \right), \tag{7}$$

in case where arm 1 is the best arm ($a^\star(P^*) = 1$). In contrast, in Theorem 4.1, we show that our strategy's upper bound is $\liminf_{T \to \infty} -\frac{1}{T} \log \mathbb{P}_{P^*} \left( \widehat{a}_T^{\mathrm{AIPW}} \neq a^\star(P^*) \right) \geq \frac{\Delta^2}{2(\sigma_1+\sigma_2)^2} - o(\Delta^2)$. The difference between our upper bounds and theirs is the existence of $-o(\Delta^2)$ term, which vanishes as $\Delta \to 0$. This difference comes from the estimation error of the variances.

**Intuitive explanation about the influence of the influence of the variance estimation.** To understand the variance estimation, we rewrite the sample average as

$$\widehat{\mu}_{a,T}^{\mathrm{SA}} = \frac{1}{T} \sum_{t=1}^{T} \frac{1}{\frac{1}{T} \sum_{t=1}^{T} \mathbb{1}[A_t^{\mathrm{KCG}} = a]} \mathbb{1}[A_t^{\mathrm{KCG}} = a]Y_t = \frac{1}{T} \sum_{t=1}^{T} \frac{1}{w_a^*} \mathbb{1}[A_t^{\mathrm{KCG}} = a]Y_t, \tag{8}$$

where we used $\sum_{t=1}^{T} \mathbb{1}[A_t^{\mathrm{KCG}} = a] = w_a^*T$. Here, we consider a strategy that estimates $w_a^*$ by estimating the variances. Let $\widetilde{w}_a$ be some estimator of $w_a^*$. Then, we design a strategy that draws arm $a$ in $\widetilde{w}_aT$ rounds in some way. In that case, the sample average roughly becomes

$$\widetilde{\mu}_{a,T}^{\mathrm{SA}} = \frac{1}{T} \sum_{t=1}^{T} \frac{1}{\frac{1}{T} \sum_{t=1}^{T} \mathbb{1}[\widetilde{A}_t = a]} \mathbb{1}[\widetilde{A}_t = a]Y_t = \frac{1}{T} \sum_{t=1}^{T} \frac{1}{\widetilde{w}_a} \mathbb{1}[\widetilde{A}_t = a]Y_t, \tag{9}$$

where $\widetilde{A}_t$ denotes an arm drawn by some strategy that draws arm $a$ in $\widetilde{w}_a T$ rounds. Then, if we recommend arm $\widetilde{a}_T := \arg\max_{a \in \{1,2\}} \widetilde{\mu}_{a,T}^{\mathrm{SA}}$ as the best arm, we evaluate

$$\liminf_{T \to \infty} -\frac{1}{T} \log \mathbb{P}_{P^*} \left( \widetilde{a}_T \neq a^\star(P^*) \right) = \liminf_{T \to \infty} -\frac{1}{T} \log \mathbb{P}_{P^*} \left( \widetilde{\mu}_{1,T}^{\mathrm{SA}} - \widetilde{\mu}_{2,T}^{\mathrm{SA}} \leq 0 \right), \tag{10}$$

From Markov's inequality, for any $\lambda > 0$, we have

$$\log \mathbb{P}_{P^*} \left( \widetilde{\mu}_{1,T}^{\mathrm{SA}} - \widetilde{\mu}_{2,T}^{\mathrm{SA}} \leq 0 \right) \leq \mathbb{E} \left[ \exp \left( T\lambda \left( \widetilde{\mu}_{1,T}^{\mathrm{SA}} - \widetilde{\mu}_{2,T}^{\mathrm{SA}} \right) \right) \right]. \tag{11}$$

To obtain the same upper bound as that in the Kaufmann et al. (2016)'s strategy, we consider the following decomposition:

$$\mathbb{E} \left[ \exp \left( T\lambda \left( \widetilde{\mu}_{1,T}^{\mathrm{SA}} - \widetilde{\mu}_{2,T}^{\mathrm{SA}} \right) \right) \right] = \mathbb{E} \left[ \exp \left( T\lambda \left( \widehat{\mu}_{1,T}^{\mathrm{SA}} - \widehat{\mu}_{2,T}^{\mathrm{SA}} \right) \right) \exp \left( T\lambda \left( \{ \widetilde{\mu}_{1,T}^{\mathrm{SA}} - \widetilde{\mu}_{2,T}^{\mathrm{SA}} \} - \{ \widehat{\mu}_{1,T}^{\mathrm{SA}} - \widehat{\mu}_{2,T}^{\mathrm{SA}} \} \right) \right) \right]. \tag{12}$$

Suppose that the following holds in some way:

$$\mathbb{E} \left[ \exp \left( T\lambda \left( \widehat{\mu}_{1,T}^{\mathrm{SA}} - \widehat{\mu}_{2,T}^{\mathrm{SA}} \right) \right) \exp \left( T\lambda \left( \{ \widetilde{\mu}_{1,T}^{\mathrm{SA}} - \widetilde{\mu}_{2,T}^{\mathrm{SA}} \} - \{ \widehat{\mu}_{1,T}^{\mathrm{SA}} - \widehat{\mu}_{2,T}^{\mathrm{SA}} \} \right) \right) \right]$$
$$= \mathbb{E} \left[ \exp \left( T\lambda \left( \widehat{\mu}_{1,T}^{\mathrm{SA}} - \widehat{\mu}_{2,T}^{\mathrm{SA}} \right) \right) \right] \mathbb{E} \left[ \exp \left( T\lambda \left( \{ \widetilde{\mu}_{1,T}^{\mathrm{SA}} - \widetilde{\mu}_{2,T}^{\mathrm{SA}} \} - \{ \widehat{\mu}_{1,T}^{\mathrm{SA}} - \widehat{\mu}_{2,T}^{\mathrm{SA}} \} \right) \right) \right].$$

Note that this decomposition does not generally hold, but we assume it since it makes it easy to understand the variance estimation problem. Under the assumption, it holds that

$$-\frac{1}{T} \log \mathbb{P}_{P^*} \left( \widetilde{a}_T \neq a^\star(P^*) \right) \tag{13}$$

$$\geq -\frac{1}{T} \log \mathbb{E} \left[ \exp \left( T\lambda \left( \widehat{\mu}_{1,T}^{\mathrm{SA}} - \widehat{\mu}_{2,T}^{\mathrm{SA}} \right) \right) \right] - \frac{1}{T} \log \mathbb{E} \left[ \exp \left( T\lambda \left( \{ \widetilde{\mu}_{1,T}^{\mathrm{SA}} - \widetilde{\mu}_{2,T}^{\mathrm{SA}} \} - \{ \widehat{\mu}_{1,T}^{\mathrm{SA}} - \widehat{\mu}_{2,T}^{\mathrm{SA}} \} \right) \right) \right] \tag{14}$$

$$\geq \frac{\Delta^2}{2(\sigma_1 + \sigma_2)^2} - \frac{1}{T} \log \mathbb{E} \left[ \exp \left( T\lambda \left( \{ \widetilde{\mu}_{1,T}^{\mathrm{SA}} - \widetilde{\mu}_{2,T}^{\mathrm{SA}} \} - \{ \widehat{\mu}_{1,T}^{\mathrm{SA}} - \widehat{\mu}_{2,T}^{\mathrm{SA}} \} \right) \right) \right]. \tag{15}$$

This inequality implies that to obtain the same upper bound as that of Kaufmann et al. (2016)'s strategy, we need to bound

$$\liminf_{T \to \infty} -\frac{1}{T} \log \mathbb{E} \left[ \exp \left( T\lambda \left( \{ \widetilde{\mu}_{1,T}^{\mathrm{SA}} - \widetilde{\mu}_{2,T}^{\mathrm{SA}} \} - \{ \widehat{\mu}_{1,T}^{\mathrm{SA}} - \widehat{\mu}_{2,T}^{\mathrm{SA}} \} \right) \right) \right] \tag{16}$$

with an arbitrage rate of convergence; more exactly, we need to show that for any $\varepsilon > 0$,

$$\liminf_{T \to \infty} -\frac{1}{T} \log \mathbb{E} \left[ \exp \left( T\lambda \left( \{ \widetilde{\mu}_{1,T}^{\mathrm{SA}} - \widetilde{\mu}_{2,T}^{\mathrm{SA}} \} - \{ \widehat{\mu}_{1,T}^{\mathrm{SA}} - \widehat{\mu}_{2,T}^{\mathrm{SA}} \} \right) \right) \right] \geq -\varepsilon$$

holds. However, it is impossible to achieve that convergence rate with commonly known theorems about convergence. Therefore, we introduced the small-gap regime, which evaluates the term as

$$\liminf_{T \to \infty} -\frac{1}{T} \log \mathbb{E} \left[ \exp \left( T\lambda \left( \{ \widetilde{\mu}_{1,T}^{\mathrm{SA}} - \widetilde{\mu}_{2,T}^{\mathrm{SA}} \} - \{ \widehat{\mu}_{1,T}^{\mathrm{SA}} - \widehat{\mu}_{2,T}^{\mathrm{SA}} \} \right) \right) \right] = -o(\Delta^2).$$

Note that this argument is not rigorous and is simplified for explanation.

### 6.3 The AIPW, IPW, and Sample Average Estimators

A key component of our analysis is the AIPW estimator, which comprises an MDS and boasts minimum asymptotic variance. By using the properties of an MDS, we tackle the dependence among observations. The upper bound can also be applied to the Inverse Probability Weighting (IPW) estimator, but in this case, the upper bound may not coincide with the lower bound. This discrepancy occurs because the AIPW estimator's asymptotic variance is smaller than the IPW estimator's. The minimum variance property of the AIPW estimator stems from the efficient influence function (Hahn, 1998; Tsiatis, 2007).

We conjecture that the asymptotic optimality of strategies employing the naive sample average estimator in the recommendation rule can be demonstrated, although we do not prove it in this study. This is because

Hahn et al. (2011) shows that, using the CLT, the AIPW and sample average estimators have the same asymptotic distribution. However, due to the inability to utilize MDS properties and the presence of sample dependency, the analysis becomes challenging when we derive a corresponding result for a large deviation (exponential rate of the probability of misidentification).

For the reader's reference, we detail the problems related to the IPW estimator and the sample average estimator.

**The NA-IPW strategy.** We consider the following strategy. In the NA-AIPW strategy, instead of the AIPW estimator, we use the following IPW estimator to estimate the means:

$$\widehat{\mu}_{a,T}^{\mathrm{IPW}} := \frac{1}{T} \sum_{t=1}^{T} \psi_{a,t}^{\mathrm{IPW}}, \qquad \text{where } \psi_{a,t}^{\mathrm{IPW}} := \frac{\mathbb{1}[A_t = a]Y_{a,t}}{\widehat{w}_{a,t}}. \tag{17}$$

At the end of the experiment (after the round $t = T$), we recommend $\widehat{a}_T^{\mathrm{IPW}}$ as

$$\widehat{a}_T^{\mathrm{IPW}} := \begin{cases} 1 & \text{if } \widehat{\mu}_{1,T}^{\mathrm{IPW}} \geq \widehat{\mu}_{2,T}^{\mathrm{IPW}}, \\ 2 & \text{otherwise.} \end{cases} \tag{18}$$

We refer to this strategy as the NA-IPW strategy, whose probability of misidentification of this strategy is given as follows.

**Theorem 6.1** (Upper bound of the NA-IPW strategy). *For any $P^* \in \mathcal{P}^{\mathrm{G}}$, the following holds as $\Delta \to 0$:*

$$\liminf_{T \to \infty} -\frac{1}{T} \log \mathbb{P}_{P^*} \left( \widehat{a}_T^{\mathrm{IPW}} \neq a^\star(P^*) \right) \geq \frac{\Delta^2}{2(\sigma_1 + \sigma_2)\left( \frac{\zeta_1^*}{\sigma_1} + \frac{\zeta_2^*}{\sigma_2} \right)} - o\left(\Delta^2\right),$$

*where $\zeta_a^* := \mathbb{E}_{P^*}\left[ Y_{a,t}^2 \right]$.*

*Proof.* Let us define $V^{\mathrm{IPW}} := \frac{\zeta_1^*}{w_1^*} + \frac{\zeta_2^*}{w_2^*} - \Delta^2$ and $\Psi^{\mathrm{IPW}} := \left\{ \psi_{1,t}^{\mathrm{IPW}} - \psi_{2,t}^{\mathrm{IPW}} - \Delta \right\} / V^{\mathrm{IPW}}$. Then, we have

$$\mathbb{E}_{P^*}\left[ \left( \Psi_t^{\mathrm{IPW}} \right)^2 |\mathcal{F}_{t-1} \right] = \mathbb{E}_{P^*}\left[ \left( \psi_{1,t}^{\mathrm{IPW}} - \psi_{2,t}^{\mathrm{IPW}} - \Delta \right)^2 \Big| \mathcal{F}_{t-1} \right]$$

$$= \mathbb{E}_{P^*}\left[ \left( \frac{\mathbb{1}[A_t = 1]Y_{1,t}}{\widehat{w}_{1,t}} - \frac{\mathbb{1}[A_t = 2]Y_{2,t}}{\widehat{w}_{2,t}} - \Delta \right)^2 |\mathcal{F}_{t-1} \right]$$

$$= \mathbb{E}_{P^*}\left[ \left( \frac{\mathbb{1}[A_t = 1]Y_{1,t}}{\widehat{w}_{1,t}} - \frac{\mathbb{1}[A_t = 2]Y_{2,t}}{\widehat{w}_{2,t}} \right)^2 |\mathcal{F}_{t-1} \right] - \Delta^2$$

$$= \mathbb{E}_{P^*}\left[ \frac{Y_{1,t}^2}{\widehat{w}_{1,t}} - \frac{Y_{2,t}^2}{\widehat{w}_{2,t}} |\mathcal{F}_{t-1} \right] - \Delta^2 \to V^{\mathrm{IPW}}$$

as $T \to \infty$. By replacing $V$ and $\Psi$ in the proof of Theorem 4.1 with $V^{\mathrm{IPW}}$ and $\Psi^{\mathrm{IPW}}$, we obtain

$$\liminf_{T \to \infty} -\frac{1}{T} \log \mathbb{P}_{P^*} \left( \widehat{a}_T^{\mathrm{IPW}} \neq a^\star(P^*) \right) \geq \frac{\Delta^2}{2\left( \frac{\zeta_1^*}{w_1^*} + \frac{\zeta_2^*}{w_2^*} - \Delta^2 \right)} - o\left(\Delta^2\right),$$

where the RHS is equal to $\frac{\Delta^2}{2\left( \frac{\zeta_1^*}{w_1^*} + \frac{\zeta_2^*}{w_2^*} \right)} - o\left(\Delta^2\right)$. The proof is complete. $\qquad\square$

Note that $2(\sigma_1 + \sigma_2)\left( \frac{\zeta_1^*}{\sigma_1} + \frac{\zeta_2^*}{\sigma_2} \right) \geq 2(\sigma_1 + \sigma_2)^2$ since $\left( \frac{\zeta_1^*}{\sigma_1} + \frac{\zeta_2^*}{\sigma_2} \right) \geq \left( \frac{\zeta_1^* - (\mu_1^*)^2}{\sigma_1} + \frac{\zeta_2^* - (\mu_2^*)^2}{\sigma_2} \right) = (\sigma_1 + \sigma_2)$. Therefore, the upper bound of probability of misidentification of the NA-IPW strategy is larger than that of the NA-AIPW strategy (Note that the inequality is flipped due to $-\frac{1}{T} \log \mathbb{P}_{P^*}$; that is, $\frac{\Delta^2}{2(\sigma_1 + \sigma_2)^2} \geq$

$\frac{\Delta^2}{2(\sigma_1+\sigma_2)\left(\frac{\varsigma_1^*}{\sigma_1}+\frac{\varsigma_2^*}{\sigma_2}\right)}$ implies that the upper bound of the probability of misidentification of the NA-AIPW strategy is smaller than that of the NA-IPW strategy). In the case of the evaluation using the CLT, similar results have been known in existing studies, such as Hirano et al. (2003) and Kato et al. (2020).

**The NA-SA strategy.** Next, we consider the following strategy. In the NA-AIPW strategy, instead of the AIPW estimator, we use the following sample average estimator:

$$\widehat{\mu}_{a,T}^{\mathrm{SA}} := \frac{1}{\sum_{t=1}^{T} \mathbb{1}[A_t = a]} \sum_{t=1}^{T} \mathbb{1}[A_t = a]Y_t = \frac{1}{T}\sum_{t=1}^{T} \psi_{a,t}^{\mathrm{SA}}, \qquad \text{where } \psi_{a,t}^{\mathrm{SA}} := \frac{\mathbb{1}[A_t = a]Y_t}{\frac{1}{T}\sum_{t=1}^{T}\mathbb{1}[A_t = a]}. \tag{19}$$

At the end of the experiment (after the round $t = T$), we recommend $\widehat{a}_T^{\mathrm{IPW}}$ as

$$\widehat{a}_T^{\mathrm{SA}} := \begin{cases} 1 & \text{if } \widehat{\mu}_{1,T}^{\mathrm{SA}} \geq \widehat{\mu}_{2,T}^{\mathrm{SA}}, \\ 2 & \text{otherwise.} \end{cases} \tag{20}$$

We refer to this strategy as the NA-SA strategy. Evaluation of the probability of misidentification of this strategy is not easy since we cannot employ a martingale property, which has been used in the analysis of the NA-AIPW strategy and the NA-IPW strategy. In order to derive its upper bound, we need to evaluate $\widehat{\mu}_{1,T}^{\mathrm{SA}} - \widehat{\mu}_{2,T}^{\mathrm{SA}}$. Here, note that

$$\widehat{\mu}_{1,T}^{\mathrm{SA}} - \widehat{\mu}_{2,T}^{\mathrm{SA}} = \widehat{\mu}_{1,T}^{\mathrm{SA}} - \widehat{\mu}_{2,T}^{\mathrm{SA}} - \left\{ \frac{\mathbb{1}[A_t = 1]\left(Y_{1,t} - \mu_1^*\right)}{\widehat{w}_{1,t}} - \frac{\mathbb{1}[A_t = 2]\left(Y_{2,t} - \mu_2^*\right)}{\widehat{w}_{2,t}} - \Delta \right\} \tag{21}$$

$$+ \left\{ \frac{\mathbb{1}[A_t = 1]\left(Y_{1,t} - \mu_1^*\right)}{\widehat{w}_{1,t}} - \frac{\mathbb{1}[A_t = 2]\left(Y_{2,t} - \mu_2^*\right)}{\widehat{w}_{2,t}} - \Delta \right\} \tag{22}$$

holds, and the variance of $\Psi_t^* := \left\{ \frac{\mathbb{1}[A_t=1]\left(Y_{1,t}-\mu_1^*\right)}{\widehat{w}_{1,t}} - \frac{\mathbb{1}[A_t=2]\left(Y_{2,t}-\mu_2^*\right)}{\widehat{w}_{2,t}} - \Delta \right\}$ is $V$ in the proof of Theorem 4.1, and $\{\Psi_t^*\}_{t=1}^{T}$ consists of an MDS. Therefore, if $\widehat{\mu}_{1,T}^{\mathrm{SA}} - \widehat{\mu}_{2,T}^{\mathrm{SA}} - \left\{ \frac{\mathbb{1}[A_t=1]\left(Y_{1,t}-\mu_1^*\right)}{\widehat{w}_{1,t}} - \frac{\mathbb{1}[A_t=2]\left(Y_{2,t}-\mu_2^*\right)}{\widehat{w}_{2,t}} - \Delta \right\} = 0$, we can directly apply the proof of Theorem 4.1 to obtain the same upper bound in Theorem 4.1. However, $\widehat{\mu}_{1,T}^{\mathrm{SA}} - \widehat{\mu}_{2,T}^{\mathrm{SA}} - \left\{ \frac{\mathbb{1}[A_t=1]\left(Y_{1,t}-\mu_1^*\right)}{\widehat{w}_{1,t}} - \frac{\mathbb{1}[A_t=2]\left(Y_{2,t}-\mu_2^*\right)}{\widehat{w}_{2,t}} - \Delta \right\}$ is not zero and remains as a bias term, and it is known that its evaluation requires several techniques. For example, to show $\sqrt{T}\left(\widehat{\mu}_{1,T}^{\mathrm{SA}} - \widehat{\mu}_{2,T}^{\mathrm{SA}} - \Delta\right) \xrightarrow{\mathrm{d}} \mathcal{N}(0, V)$, Hahn et al. (2011) bounds $\widehat{\mu}_{1,T}^{\mathrm{SA}} - \widehat{\mu}_{2,T}^{\mathrm{SA}} - \left\{ \frac{\mathbb{1}[A_t=1]\left(Y_{1,t}-\mu_1^*\right)}{\widehat{w}_{1,t}} - \frac{\mathbb{1}[A_t=2]\left(Y_{2,t}-\mu_2^*\right)}{\widehat{w}_{2,t}} - \Delta \right\}$ using the property of the stochastic equicontinuity, which is based on the arguments in Hirano et al. (2003). This problem is related to the use of the Donsker condition in semiparametric analysis, as explained in Kennedy (2016). We may show the upper bound of the NA-SA strategy by using a similar approach used in Hahn et al. (2011), but there are two issues. First, it is unknown what condition corresponds to the stochastic equicontinuity in the setting of BAI, where the samples are dependent. Second, it is unclear whether we can directly apply the stochastic equicontinuity or similar properties to show the large deviation upper bound since such conditions have been used for the central limit evaluation. Therefore, although the findings of Hirano et al. (2003) and Hahn et al. (2011) may aid in resolving this issue, it is an open issue how we use it. Note that this issue caused by the bias of $\widehat{\mu}_{1,T}^{\mathrm{SA}} - \widehat{\mu}_{2,T}^{\mathrm{SA}}$, which is non-zero. Also note that in contrast, the bias of $\widehat{\mu}_{1,T}^{\mathrm{AIPW}} - \widehat{\mu}_{2,T}^{\mathrm{AIPW}}$ is zero due to the properties of an MDS.

## 6.4 The Tracking Strategy

In fixed-confidence BAI, the tracking strategy is popular, as used in Garivier & Kaufmann (2016). In the existing studies of the Neyman allocation, such a strategy has been used. For example, Hahn et al. (2011) splits the whole samples into two groups. In the first stage, we uniformly randomly draw each arm and

estimate $w_a^*$. In the second stage, for the estimators $\widehat{w}_a$ of $w_a^*$ we draw each arm so that $\frac{1}{T}\sum_{t=1}^{T}\mathbb{1}[A_t = a] = \widehat{w}_a$ holds. Then, Hahn et al. (2011) estimates $\Delta = \mu_1^* - \mu_2^*$ using the sample average estimator. This strategy is quite similar to that in Garivier & Kaufmann (2016), since it draws arms to track the ratio of $w_a^*$.

However, the strategy of Hahn et al. (2011) makes analyzing upper bounds difficult. As we explained in Section 6.3, in our analysis, the unbiasedness of the AIPW estimator plays an important role. In contrast, if we use the tracking strategy, we cannot employ the property of $\mathbb{E}_{P^*}\left[\frac{\mathbb{1}[A_t=a]}{\widehat{w}_t}|\mathcal{F}_{t-1}\right] = 1$. Note that the NA-AIPW strategy draws arm $A_t$ with probability $\widehat{w}_t$, but the tracking strategy draws arm $A_t$ more complicatedly, under which we cannot use the martingale property.

As well as we explained in Section 6.3, the bias term makes the analysis significantly difficult. According to the existing studies, we need to use some techniques for the analysis, such as the Donsker condition (Hirano et al., 2003; Hahn et al., 2011). Existing studies have proposed using the AIPW estimator to avoid this issue, as shown in van der Laan (2008) and Kato et al. (2020).

Thus, although we acknowledge the possibility of using the tracking strategy, the analysis requires some sophisticated techniques. We expect that existing studies such as (Hirano et al., 2003) and Hahn et al. (2011) will help the analysis, but it is still an open issue. Note that even in the tracking strategy, existing strategy such as (Hirano et al., 2003) and Hahn et al. (2011) bypass the evaluation of the AIPW-type estimators in the analysis. This proof procedure is also related to the semiparmaetric efficiency bound (Hahn, 1998), under which the semiparametric efficient score is given as $\Psi_t^* = \left\{ \frac{\mathbb{1}[A_t=1]\left(Y_{1,t}-\mu_1^*\right)}{\widehat{w}_{1,t}} - \frac{\mathbb{1}[A_t=2]\left(Y_{2,t}-\mu_2^*\right)}{\widehat{w}_{2,t}} - \Delta \right\}$.

# 7 Related Work

This section presents related works.

## 7.1 On the Asymptotic Optimality in Fixed-Budget BAI

There is a long debate on the optimal strategies for fixed-budget BAI. Glynn & Juneja (2004) develops their strategies by using the large deviation principles. However, while they justify their strategies using the large deviation principles, they do not provide lower bounds for strategies. Therefore, there remains a question about whether their strategies are truly asymptotically optimal.

Kaufmann et al. (2016) establishes distribution-dependent lower bounds for BAI with fixed confidence and budget, utilizing change-of-measure arguments. According to their results, we can confirm that for two-armed Gaussian bandits, the strategy of Glynn & Juneja (2004) is optimal.

However, Kaufmann et al. (2016) leaves lower bounds for multi-armed fixed-budget BAI as an open issue. Based on the arguments of Glynn & Juneja (2004) and Russo (2020), Kasy & Sautmann (2021) attempts to derive an asymptotically optimal strategy, but their attempt does not succeed. As pointed out by Ariu et al. (2021), without additional assumptions, there exists an instance $P^*$ whose lower bound is larger than that of Kaufmann et al. (2016). This result is based on another lower bound discovered by Carpentier & Locatelli (2016). These arguments are summarized by Qin (2022).

To address this issue, Kato et al. (2023b) and Degenne (2023) consider a restriction such that sampling rules do not depend on $P^*$. Under this restriction, we can show the asymptotic optimality of the strategy provided by Glynn & Juneja (2004), which requires full knowledge about $P^*$ and is practically infeasible.

Komiyama et al. (2022) and Atsidakou et al. (2023) discuss asymptotically optimal strategies from minimax and Bayesian perspectives, respectively, where the leading factor ignoring some constant terms of lower and upper bounds match, unlike our optimality up to constant terms. This open issue is further explored by Komiyama et al. (2022), Wang et al. (2023a), Wang et al. (2023b), and Kato (2023).

Note that in the fixed confidence BAI setting, Garivier & Kaufmann (2016) proposes a strategy with an upper bound matching the derived lower bound. However, in the fixed-budget BAI, it remains unclear whether a strategy with an upper bound matching Kaufmann et al. (2016)'s lower bound exists.

Alternative lower bounds have been proposed by Audibert et al. (2010), Bubeck et al. (2011), Komiyama et al. (2023) and Kato et al. (2023a) for the expected simple regret minimization, which is another performance measure different from the probability of misidentification.

Some research employs local asymptotics to examine the asymptotic optimality of the Neyman allocation rule in this context, such as Armstrong (2022) and Adusumilli (2022).

Ordinal optimization in the operations research community is another related field (Ahn et al., 2021; Chen et al., 2000).

### 7.2 Extension to BAI in Multi-Armed Bandit (MAB) Problems

In contrast to two-armed bandit problems and BAI with fixed confidence, lower bounds for MAB problems remain unknown. One primary reason is the reversal of KL divergence. Kato et al. (2023b), Degenne (2023), Kato (2023) consider strategies that use sampling rules that are (asymptotically) invariant for any $P^* \in \mathcal{P}^{\mathrm{G}}$. Such a class of strategies is sometimes called *static* in the sense that it cannot estimate parameters during an adaptive experiment to avoid the dependency on $P^*$. However, if we consider Gaussian bandit models, sampling strategies that are invariant for $P^*$ do not imply non-adaptive (static) strategies because we can still adaptively estimate the variances during an adaptive experiment (the variances are assumed to be the same for any $P^*$).

## 8 Simulation Studies

This section provides simulation studies to investigate the empirical performance of the NA-AIPW strategy. For comparison, we also investigate the performances of the NA-IPW and NA-SA strategies defined in Section 6.3. Furthermore, we also conduct simulation studies of the "oracle" strategy with the known variances, denoted by Oracle, and the uniform strategy that draws an equal number of arms, denoted by Uniform. We recommend an arm with the highest sample average in the Oracle and Uniform strategies.[2]

Throughout the experiment, we set arm 1 as the best arm. We conduct experiments with the three settings.

In the first experiment, we set $\mu_1^* = 1.00$ and choose $\mu_2^*$ from the set $\{0.80, 0.85, 0.90, 0.95, 0.99\}$. The variances $(\sigma_1, \sigma_2)$ are selected with a probability of $1/2$ from either $(1, v_2)$ or $(v_2, 1)$, where $v_2$ is chosen from $5, 10, 20, 50$. We continue the strategies until $T = 10,000$ and report the empirical probability of misidentification at $T \in \{100, 200, 300, \dots, 9,900, 10,000\}$.[3] We conduct $1,000$ independent trials for each choice of parameters and plot the results in Figures 1 and 2.

In the second experiment, the variances $(\sigma_1, \sigma_2)$ are selected with a probability of $1/2$ from either $(5, v_2)$ or $(v_2, 5)$, where $v_2$ is chosen from $5, 10, 20, 50$. The other settings are the same as the first experiment. The results are shown in Figures 3 and 4.

In the third experiment, we set $\mu_1^* = 10.00$ and choose $\mu_2^*$ from the set $\{9.80, 9.85, 9.90, 9.95, 9.99\}$. The other settings are the same as the first experiment. The results are shown in Figures 5 and 6.

Our theoretical results imply that the probability of misidentifications of the NA-AIPW and Oracle strategies approach the same as $\Delta \to 0$. We can confirm the phenomenon. In the results, the Oracle strategy is a bit better than the NA-AIPW strategy when $\Delta$ is large. However, the gap approaches zero as $\Delta \to 0$. Note that when $\Delta$ is large, the convergence of the probability of misidentification is very fast, so the gap is still not so large even if $\Delta$ is large because both the probability of misidentifications of the NA-AIPW and Oracle strategies converge to zero very fast.

---

[2]The oracle strategy is the one proposed by Glynn & Juneja (2004) and Kaufmann et al. (2016). The Uniform strategy with the recommendation rule is referred to as the Uniform-Empirical Best Arm (EBA) strategy by Bubeck et al. (2011).

[3]This means we report the empirical probability of misidentification at $T \in \{100, 200, 300, \dots, 9,900, 10,000\}$.

We can also find that the performance improvement of the NA-AIPW strategy from the Uniform strategy is large as the difference of variances is large. For example, in the second experiment, when $(\sigma_1, \sigma_2) = (5.5)$, there is no improvement by using the NA-AIPW strategy from the Uniform strategy because the NA allocation also leads us to draw each arm with equal ratio.

The difference between the NA-AIPW and NA-IPW strategies becomes large as the mean outcome of each arm becomes large. We can find that in the third setting, the NA-IPW strategy behaves badly since $\mu_1^* = 10.00$ and $\mu_2^*$ is chosen from $\{9.80, 9.85, 9.90, 9.95, 9.99\}$, while $\mu_1^* = 1.00$ and $\mu_2^*$ is chosen from $\{0.80, 0.85, 0.90, 0.95, 0.99\}$ in the first and second settings.

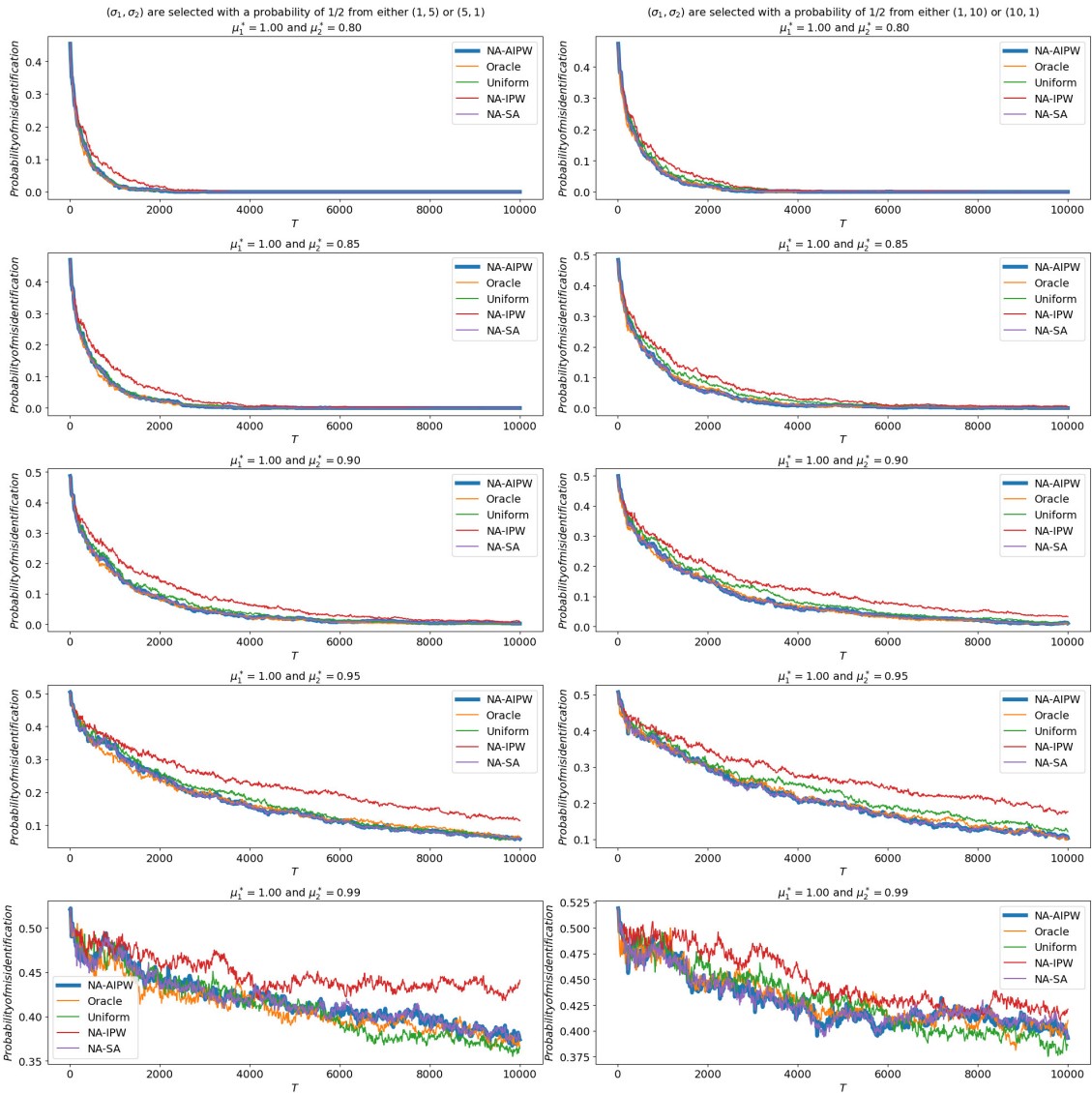

Figure 1: The results are under the first setting. We set $\mu_1^* = 1.00$ and choose $\mu_2^*$ from the set $\{0.80, 0.85, 0.90, 0.95, 0.99\}$. The variances $(\sigma_1, \sigma_2)$ are selected with a probability of $1/2$ from either $(1, v_2)$ or $(v_2, 1)$, where $v_2$ is chosen from $5, 10$. We conduct $1,000$ independent trials and report the empirical probability of misidentification at $T \in \{100, 200, 300, \ldots, 9,900, 10,000\}$.

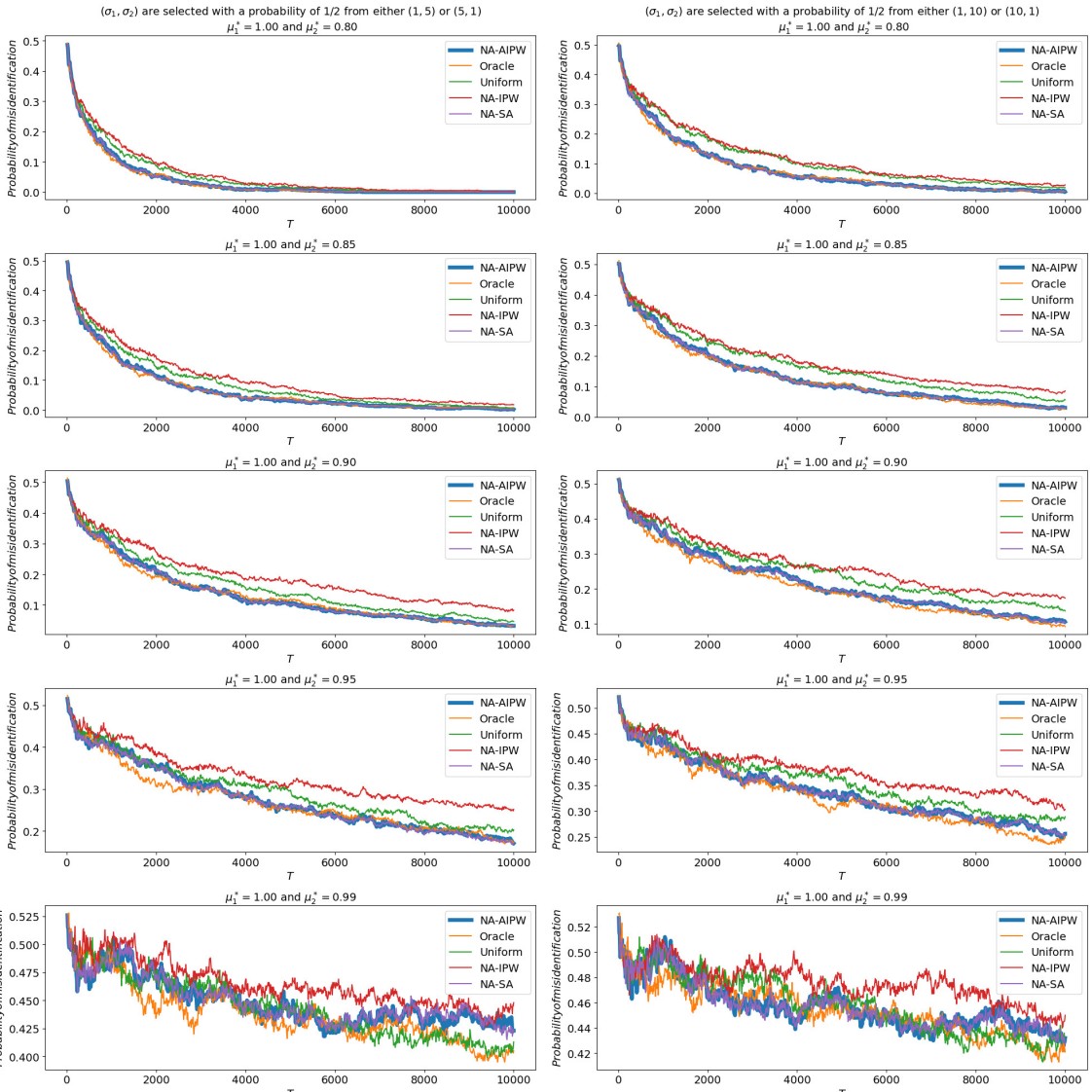

Figure 2: The results are under the first setting. We set $\mu_1^* = 1.00$ and choose $\mu_2^*$ from the set $\{0.80, 0.85, 0.90, 0.95, 0.99\}$. The variances $(\sigma_1, \sigma_2)$ are selected with a probability of $1/2$ from either $(1, v_2)$ or $(v_2, 1)$, where $v_2$ is chosen from $20, 50$. We conduct $1,000$ independent trials and report the empirical probability of misidentification at $T \in \{100, 200, 300, \ldots, 9,900, 10,000\}$.

# 9    Conclusion

This study investigated fixed-budget BAI for two-armed Gaussian bandits with unknown variances. We first reviewed the lower bound shown by Kaufmann et al. (2016). Then, we proposed the NA-AIPW strategy and found that its probability of misidentification matches the lower bound when the budget approaches infinity and the gap between the expected rewards of the two arms approaches zero. We referred to this setting as the small-gap regime and the optimality as the local asymptotic optimality. Although there are several remaining open questions, our result provides insight into long-standing open problems in BAI.

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

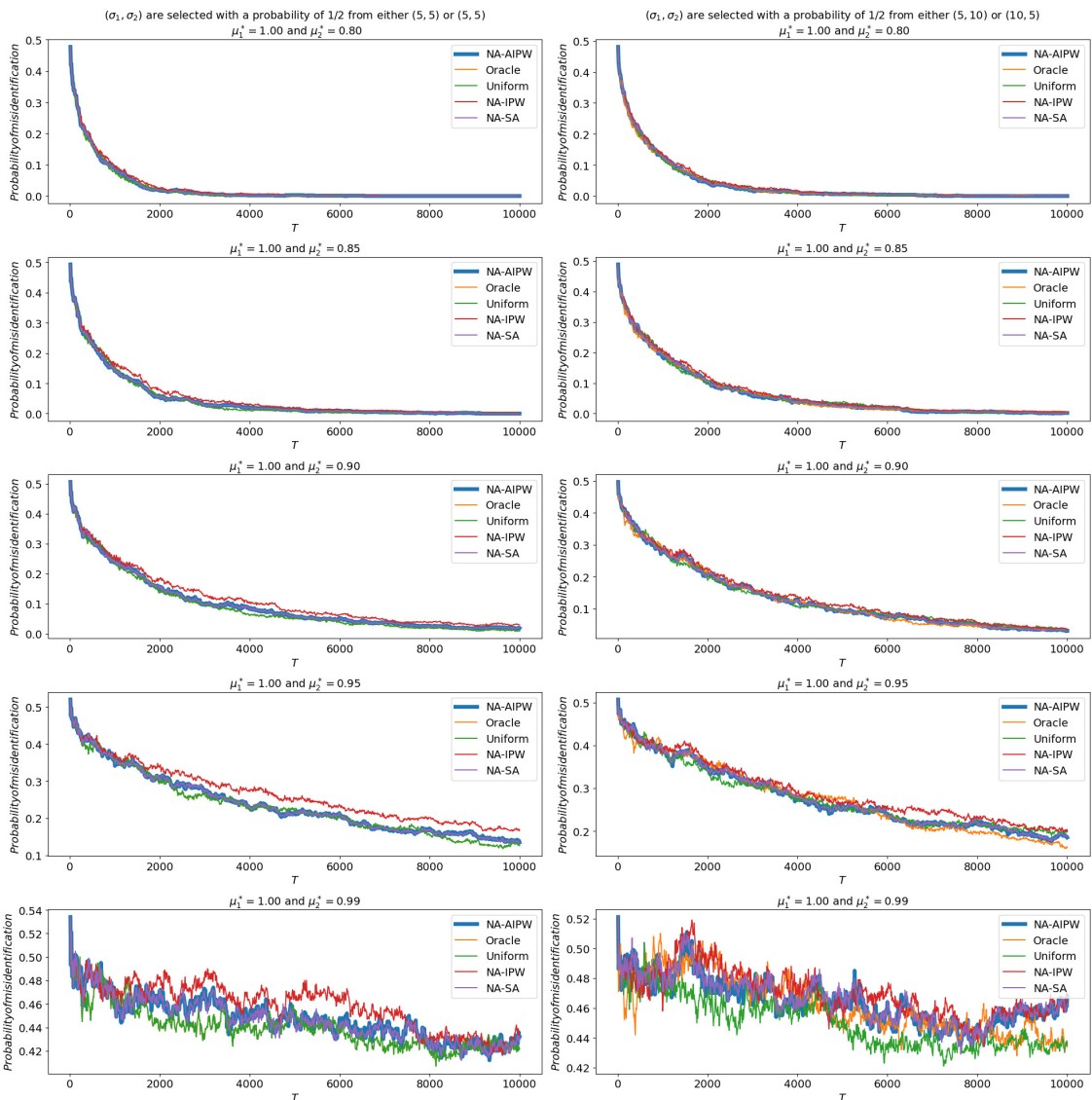

Figure 3: The results are under the first setting. We set $\mu_1^* = 1.00$ and choose $\mu_2^*$ from the set $\{0.80, 0.85, 0.90, 0.95, 0.99\}$. The variances $(\sigma_1, \sigma_2)$ are selected with a probability of $1/2$ from either $(5, v_2)$ or $(v_2, 5)$, where $v_2$ is chosen from $5, 10$. We conduct $1,000$ independent trials and report the empirical probability of misidentification at $T \in \{100, 200, 300, \ldots, 9,900, 10,000\}$.

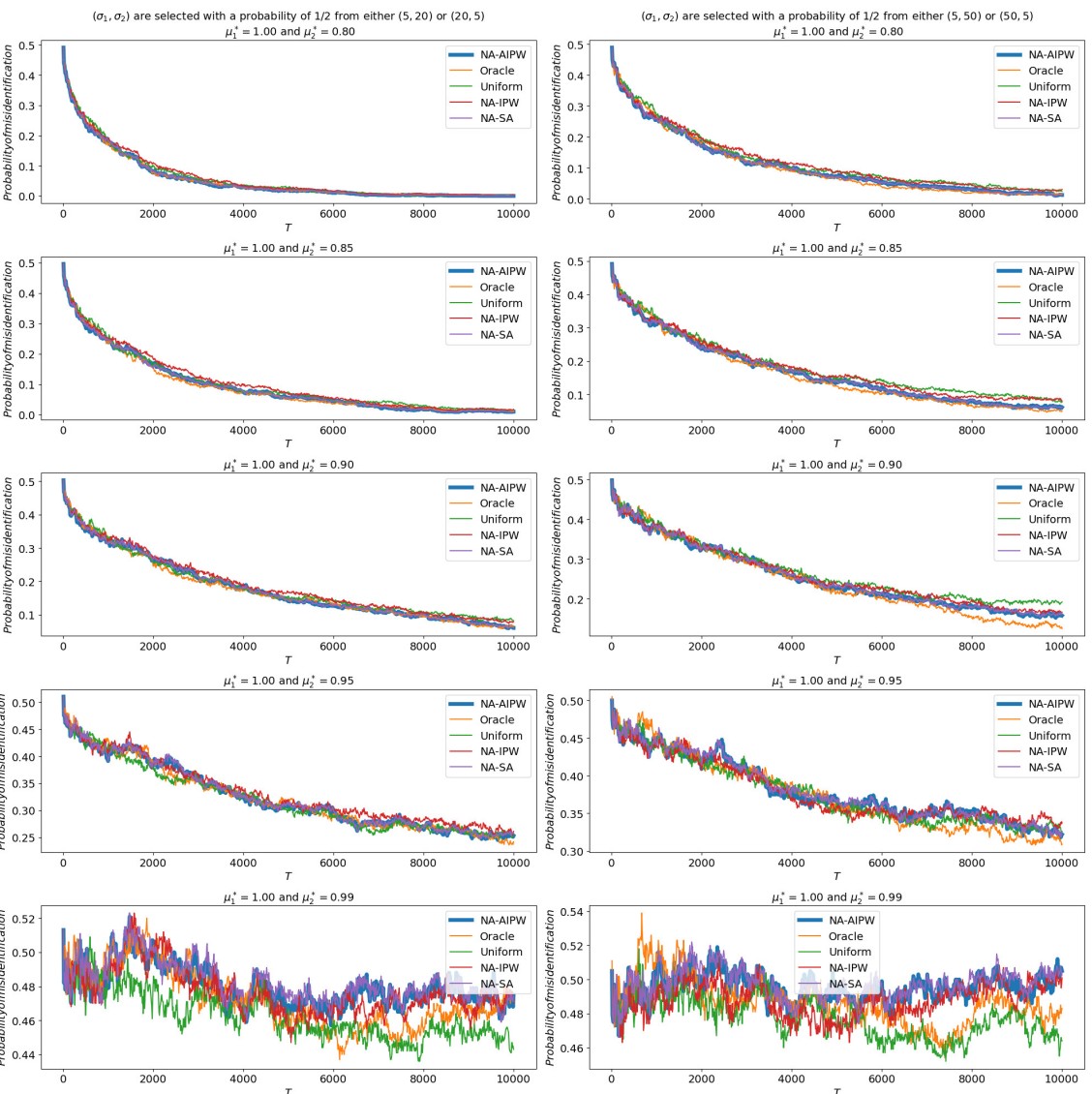

Figure 4: The results are under the first setting. We set $\mu_1^* = 1.00$ and choose $\mu_2^*$ from the set $\{0.80, 0.85, 0.90, 0.95, 0.99\}$. The variances $(\sigma_1, \sigma_2)$ are selected with a probability of $1/2$ from either $(5, v_2)$ or $(v_2, 5)$, where $v_2$ is chosen from $20, 50$. We conduct $1,000$ independent trials and report the empirical probability of misidentification at $T \in \{100, 200, 300, \ldots, 9,900, 10,000\}$.

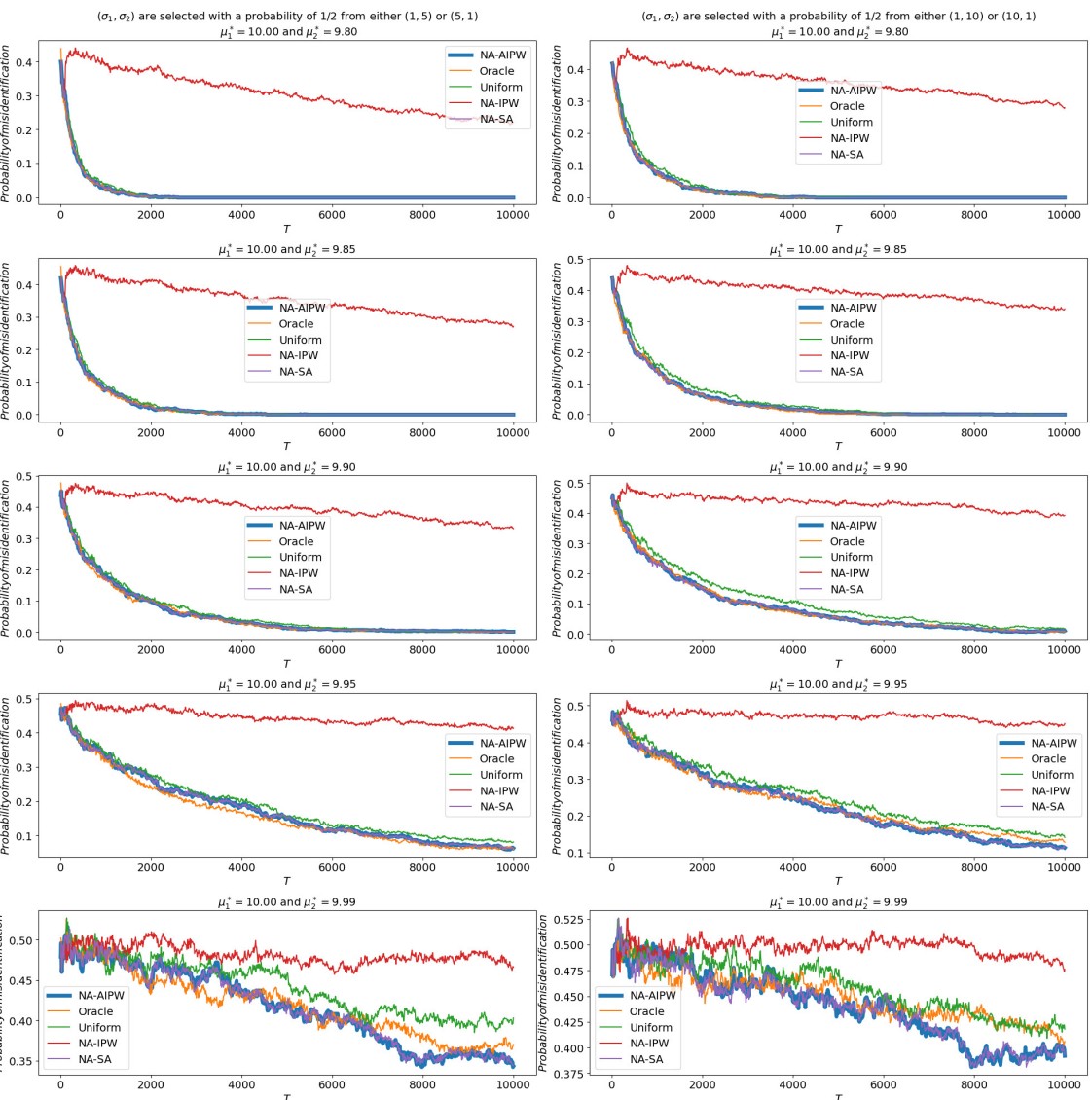

Figure 5: The results are under the first setting. We set $\mu_1^* = 10.00$ and choose $\mu_2^*$ from the set $\{9.80, 9.85, 9.90, 9.95, 9.99\}$. The variances $(\sigma_1, \sigma_2)$ are selected with a probability of $1/2$ from either $(1, v_2)$ or $(v_2, 1)$, where $v_2$ is chosen from $5, 10$. We conduct $1,000$ independent trials and report the empirical probability of misidentification at $T \in \{100, 200, 300, \ldots, 9,900, 10,000\}$.

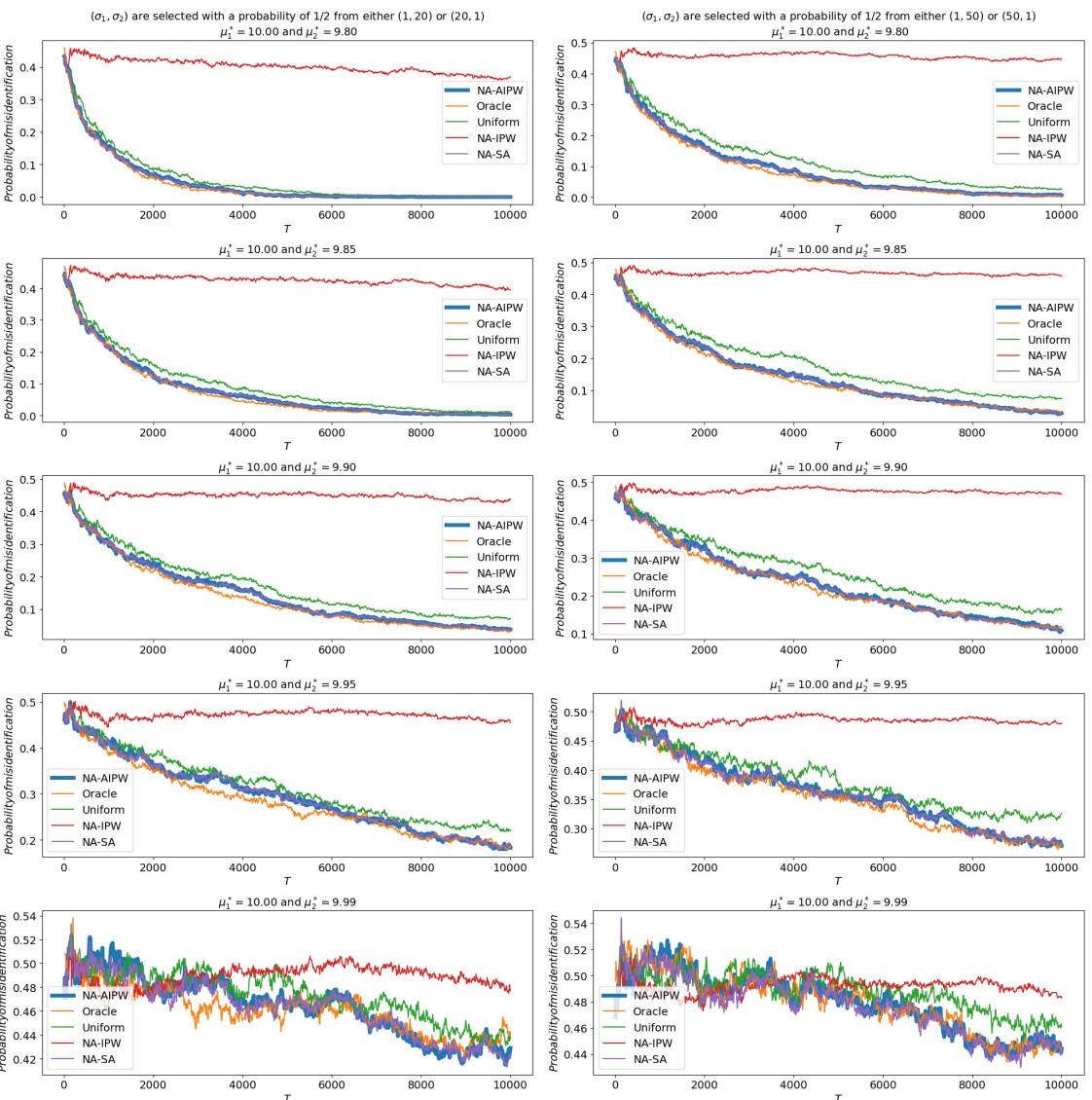

Figure 6: The results are under the first setting. We set $\mu_1^* = 10.00$ and choose $\mu_2^*$ from the set $\{9.80, 9.85, 9.90, 9.95, 9.99\}$. The variances $(\sigma_1, \sigma_2)$ are selected with a probability of $1/2$ from either $(1, v_2)$ or $(v_2, 1)$, where $v_2$ is chosen from $20, 50$. We conduct $1,000$ independent trials and report the empirical probability of misidentification at $T \in \{100, 200, 300, \ldots, 9,900, 10,000\}$.

