# OpenReview forum: "Locally Optimal Fixed-Budget Best Arm Identification in Two-Armed Gaussian Bandits with Unknown Variances"
_TMLR — Rejected by TMLR_

### Review · Reviewer_jcHp · 2024-02-08

**Summary Of Contributions:**

This paper considers the problem of BAI with fixed budget and two arms with Gaussian distributions. The main difficulty wrt previous works is that the arms might have different variances, which are unknown to the learner.
This work proposes an asymptotically (both in $T\to\infty$ and $\Delta\to 0$) optimal algorithm, based on empirical estimates of the variance.

**Audience:**

Yes

**Claims And Evidence:**

No

**Requested Changes:**

I would like the authors to change the algorithm and proof, so that it
1) is clearer,
2) is correct, i.e., with a proved Lemma 5.1 and more rigorously quantified $O$ notations
3) does not need a prior knowledge of (an upper bound of) the variances,
4) is optimal even in the large gap regime.

Moreover, I would like the authors to make the paper clearer, more precise in general. As some examples: the function thre is not defined. Actually I believe it is just a max, so it might be needed to introduce some weird notation. Another example is the word MDS that is used, without any reference to its meaning (Martingale Difference Sequence).
Another example is the paragraph on the workds of Airu et al. (2021), Qin (2022) and Degenne (2023). I only had a look at Degenne's work, but it seems to only claim *non existence of optimal strategies* for K>2 arms (in Gaussian BAI), so this would be the reason for the absence of contradiction?

**Strengths And Weaknesses:**

I have major concerns regarding the rigorousness of the presented results, and even believe that a much simpler algorithm (and analysis) can lead to optimal upper bounds even in the large gap regime (i.e. $\Delta$ not arbitrarily close to $0$). As an example, I think the current algorithm is unnecessarily complex through the use of AIPW recommendation (instead of recommending the best empirical arm).

First, I find the paper poorly written. There are many confusing, inaccurate paragraphs as well as many typos. Regardless of the comprehensive aspect of the paper, this also translates in the mathematical statements. It thus even has consequences on the mathematical rigor of the presented results.

My major such concern is about the use of unprecised constants $C_{\mu},C_{\sigma^2}$ in the algorithm. Given the used estimate of the variance, I believe the algorithm requires $C_{\sigma^2}\geq \sigma^2$ to be optimal. Yet, no mention of the required values of those constants is done in Theorem 4.1. Actually, Lemma 5.1 should depend on this condition ($C_{\sigma^2}\geq \sigma^2$). This is of major importance, as it requires the algorithm to be aware of a *positive* upper bound of the variances before hand. This does not make the proposed algorithm very adaptative to the ignorance of the variances...

Actually, a proof of Lemma 5.1 is missing, and I believe it is currently wrong. Even with the condition ($C_{\sigma^2}\geq \sigma^2$), it is not obvious that every arm will be pull an infinite number of time. Indeed, the empirical variance of an arm can be arbitrarily close to $0$ at some point (luckily, it cannot be $0$ as we consider continuous distributions). In that case, it can take an arbitrarily large time to be pulled again. As a consequence, it is even unclear that an arm will be pulled an infinite number of times almost surely. I propose an easy fix for that further below.

The main theorem (4.1) is poorly stated: what are the quantities hidden in the $O(\Delta^3)$. I believe the constants in this $O$ would also depend on $\sigma_1,\sigma_2$, meaning that the result actually only holds *for fixed $\sigma_1,\sigma_2$*, while $\Delta$ converges to $0$.

I find the proofs unnecessarily complex: they could be much further simplified. As an example, Lemma 5.4 is just a Cesaro convergence following Lemma 5.3 (it should be a one line proof).

When looking at the Step 5 of the proof, it actually even seems that the $O(\Delta^3)$ term could be removed from Theorem 4.1. Indeed, in that step, a quantity is bounded in $O(\frac{\lambda^3}{\sqrt{T}})$ and is then replaced by $O(\lambda^3)$. We could instead just keep $O(\frac{\lambda^3}{\sqrt{T}})$ in the next lines. Once we will take $T\to\infty$, this term would then simply disappears.
As a consequence, the small gap condition migth actually not be needed: the algorithm should always be optimal for large $T$ (given that Lemma 5.1 is correct).


Actually, I think that we can do a simpler algorithm, with a much simpler proof. First, a usual trick to ensure Lemma 5.1. is to force at least $\sqrt{T}$ pulls on each arm (see for example Track and Stop algorithm). In that case, each arm is pulled an infinite number of time when $T\to\infty$, but this forced exploration becomes negligible at large times. Moreover, thanks to this trick, we don't need for a constant $C_{\sigma^2}$ and we would thus have an algorithm that does not need to know an upper bound of $\sigma^2$ to actually work.
From then, instead of using the sampling rule in this paper (see $A_t$ page 4), we could simply *track* the number of pulls on each arm, so that we respect the Neyman Allocation as much as possible. In words, when not doing forced exploration, we would pull:
$$A_t = \arg\max_{k} \hat{w}_{k,t} t - N_k(t).$$
Note again that this is close to the Track and Stop design (and $t$ could be replaced by $T$ here). From then, thanks to Lemma 5.1, it easily yields that for $T$ large enough, every arm $k$ will be pulled at least $(w^*_k-\varepsilon)T$ times at the time $T$ (for a fixed $\varepsilon>0$). From there, we can simply return the best empirical arm and use the typical bounds of Kaufmann et al (2016) to show that the probability of returning the wrong arm is asymptotically optimal.

I do not see why such an algorithm would not work (with a very simple proof).

---

> ### Author Response · Authors · 2024-02-20
> **Reply to Reviewer jcHp**
>
> We are grateful for the insightful comments provided by the reviewer. Below, we address each of the concerns raised.
>
> ## 1. The existence of constants.
> ### Question 1
> > My major such concern is about the use of unprecised constants $C_{\sigma^2}$ in the algorithm. Given the used estimate of the variance, I believe the algorithm requires $C_{\sigma^2} \geq \sigma^2$ to be optimal. Yet, no mention of the required values of those constants is done in Theorem 4.1. Actually, Lemma 5.1 should depend on this condition ($C_{\sigma^2} \geq \sigma^2$). This is of major importance, as it requires the algorithm to be aware of a positive upper bound of the variances before hand. This does not make the proposed algorithm very adaptative to the ignorance of the variances...
>
> ### Answer 1
> We appreciate the reviewer's critical observation. Firstly, we believe that we require $\sigma^2 > C_{\sigma^2}$, rather than $C_{\sigma^2} \geq \sigma^2$. Acknowledging the reviewer's point, the constant $C_{\sigma^2}$ is indeed essential for technical reasons to prevent divergence of the mean estimators to infinity. The constant is just for a technical purpose, and it is enough to use a sufficiently small positive $C_{\sigma^2}$, which is known in advance of an experiment, and $\sigma^2 > C_{\sigma^2}$ holds. We have incorporated a comprehensive explanation regarding the determination of $C_{\sigma^2}$ in the revised manuscript to clarify the dependence.
>
> ## 2. Every arm will be pulled an infinite number of times
> ### Question 2
> > Actually, a proof of Lemma 5.1 is missing, and I believe it is currently wrong. Even with the condition ($C_{\sigma^2} \geq \sigma^2$), it is not obvious that every arm will be pull an infinite number of time. Indeed, the empirical variance of an arm can be arbitrarily close to $0$ at some point (luckily, it cannot be $0$ as we consider continuous distributions). In that case, it can take an arbitrarily large time to be pulled again. As a consequence, it is even unclear that an arm will be pulled an infinite number of times almost surely. I propose an easy fix for that further below.
>
> ### Answer 2
> We maintain that our manuscript assumes $\sigma^2 \geq C_{\sigma^2}$, and not the inverse. Here, $\sigma^2 \geq C_{\sigma^2}$, where $C_{\sigma^2}$ is a constant independent of $T$, ensures $w^*_a > C$ for some constant $C > 0$ independent of $T$, which also implies that each arm is indeed pulled infinitely over time. In the revised manuscript, we clarified this point.
>
> ## 3. On the $O(\lambda^3)$ term.
> ### Question 3
> > When looking at the Step 5 of the proof, it actually even seems that the $O(\Delta^3)$ term could be removed from Theorem 4.1. Indeed, in that step, a quantity is bounded in $O(\frac{\lambda^3}{\sqrt{T}})$ and is then replaced by $O(\lambda^3)$. We could instead just keep $O(\frac{\lambda^3}{\sqrt{T}})$ in the next lines. Once we will take $T\to \infty$, this term would then simply disappears. As a consequence, the small gap condition migth actually not be needed: the algorithm should always be optimal for large $T$ (given that Lemma 5.1 is correct).
>
> ### Answer 3
> The reviewer points out that we can remove $+O(\lambda^3)$ in the statement because $+O(\lambda^3/ \sqrt{T})$ converges to zero as $T\to\infty$, and thus we do not need to consider the small gap regime. However, the removal of $O(\Delta^3)$ or $-o(\Delta^2)$ is not feasible for two reasons: our initial typographical error and an oversight by the reviewer. We clarify these points as follows:
>
> - **Our typo**: Although the reviewer points out that $+O(\lambda^3/ \sqrt{T})$ converges to zero as $T\to \infty$, the$+O(\lambda^3/ \sqrt{T})$ is our typo.  The correct form is $-O(\lambda^3)$. This typo was fixed in the revised manuscript.
> - **Reviewer's oversigh**t: Thus, the $-O(\lambda^3)$ term does not vanish as $T\to \infty$. Furthermore, there is another $o(\Delta^2)$ term, which comes from the variance estimation, while the $-O(\lambda^3)$ term comes from the Taylor expansion. Even if we ignore the first  $+O(\lambda^3)$ term, the $-o(\Delta^2)$ term remains. However, the reviewer does not count the $o(\Delta^2)$ term, which requires us to consider the small gap regime $\Delta \to 0$. Otherwise, the term remains.
>
> Thus, although our typo was confusing, the main claim does not change. Furthermore, even if we ignore the $+O(\lambda^3/ \sqrt{T})$  term following the reviewer's comment (though we cannot ignore it in general), there is still the remaining $o(\Delta^2)$ term, which requires us to use the small-gap regime.

---

> > ### Author Response · Authors · 2024-02-20
> > **Reply to Reviewer jcHp (Cont.)**
> >
> > ## 4. Necessity of the small-gap regime.
> > ### Question 4
> > > As a consequence, the small gap condition migth actually not be needed: the algorithm should always be optimal for large
> >  (given that Lemma 5.1 is correct).
> >
> > ### Answer 4
> > We believe that deriving a matching upper bound is intractable without some restrictions or assumptions, such as the small-gap regime. Specifically, in the large-gap regime, aligning an upper bound with the lower bound established by Kaufman et al. (2016) would necessitate disregarding the estimation errors at ``any'' exponential rate of convergence, a scenario that is generally implausible. This challenge is elaborated upon in the newly added Section 6 of our manuscript, where we discuss the critical need for the small-gap regime due to these estimation errors.
> >
> > ## 5. The tracking strategy.
> > ### Question 5.
> > > Actually, I think that we can do a simpler algorithm, with a much simpler proof. First, a usual trick to ensure Lemma 5.1. is to force at least $\sqrt{T}$ pulls on each arm (see for example Track and Stop algorithm). In that case, each arm is pulled an infinite number of time when $T\to\infty$, but this forced exploration becomes negligible at large times. Moreover, thanks to this trick, we don't need for a constant $C_{\sigma^2}$ and we would thus have an algorithm that does not need to know an upper bound of $\sigma^2$ to actually work. From then, instead of using the sampling rule in this paper (see $A_t$ page 4), we could simply track the number of pulls on each arm, so that we respect the Neyman Allocation as much as possible. In words, when not doing forced exploration, we would pull: $A_t = \arg\max_{k} \hat{w}_{k,t} t - N_k(t)$. Note again that this is close to the Track and Stop design (and $t$ could be replaced by $T$ here). From then, thanks to Lemma 5.1, it easily yields that for $T$ large enough, every arm $k$ will be pulled at least $(w^*_k-\varepsilon)T$ times at the time  (for a fixed $\epsilon > 0$). From there, we can simply return the best empirical arm and use the typical bounds of Kaufmann et al (2016) to show that the probability of returning the wrong arm is asymptotically optimal. I do not see why such an algorithm would not work (with a very simple proof).
> >
> > ### Answer 5.
> > We appreciate this suggestion and acknowledge that we explored similar avenues over the past three years without success in achieving a matching upper bound to Kaufman et al.'s (2016) lower bound. Existing studies from the 2000s have either avoided this approach or have had to rely on certain assumptions for their proofs, as seen in works by van der Laan (2008) and Hahn et al. (2011). We identify several substantial hurdles that we believe the reviewer may have underestimated:
> >
> > - **Estimation error of the variances**: As we explained above, matching the upper bound with Kaufman et al. (2016)'s lower bound necessitates neglecting the variance estimation errors at ``any'' exponential rate, which is contrary to established probabilistic principles.
> > - **Tight variance-dependent upper bound**: Achieving a tight upper bound that accounts for variances is an additional complex challenge. When estimating the gap $\Delta$ through the central limit theorem, deriving a precise upper bound has been a longstanding issue in statistics, as highlighted by Hirano et al. (2003). Empirical process theory is often needed to derive tight bounds in CLT-based analyses.
> >
> > While the reviewer questions the complexity of our algorithm, we argue that our approach is rather simpler and more straightforward than the tracking strategy from the theoretical perspective. Our use of the AIPW estimator aligns with the ``efficient'' score function yielding the smallest variance, and we use it directly for deriving the smallest variance. For example, for estimating $\Delta$ without known variances, although Hahn et al. (2011) proposes a strategy that is similar to the one reviewer suggests, their proof also bypasses a hypothetical AIPW estimator in their proof and introduces several assumptions on variance estimators that are difficult to verify in practice.
> >
> > These points are thoroughly discussed in the newly added Section 6 of our revised manuscript.
> >
> > ## Other comments.
> > ### Question
> > > I find the proofs unnecessarily complex: they could be much further simplified. As an example, Lemma 5.4 is just a Cesaro convergence following Lemma 5.3 (it should be a one line proof).
> >
> > ### Answer
> > As the reviewer points out, the part corresponds to the almost sure convergence version of the Cesaro convergence. However, the standard Cesaro convergence does not deal with probability. Because we cannot find an appropriate reference, and existing studies such as Hadad et al. (2021) also shows the whole proof, we also add the whole proof for completeness. We explained that the result is the almost sure convergence version of the Cesaro convergence in the revised manuscript while keeping the proof for completeness.

---

> > > ### Comment · Reviewer_jcHp · 2024-02-20
> > >
> > > I thank the authors for their answer. I yet still have some doubts about this work.
> > >
> > > # Question 1
> > >
> > > Indeed, I actually missed the definition of the thre function at the end of Section 1. Now, I am wondering if there is not a typo in its definition. Following your claim, we should take $C_{\sigma^2}$ arbitrarily small. But in that case, looking at the thre function and the empirical variance estimator, the estimator should actually converge to $\frac{1}{C_{\sigma^2}}$ almost surely. In the light of this, I again reaffirm the need for a rigorous proof of Lemma 5.1.
> > >
> > > Actually, this observation makes me even doubt on the empirical success of such algorithm. I would thus now join the other reviewers on the need for some numerical experiment.
> > >
> > > # Question 2
> > >
> > > Even if Lemma 5.1 was correct in its current form, I still think that using a forced exploration of order $\sqrt{T}$ would not change anything about the current analysis, while making it available without any knowledge of a lower (or upper) bound of the variances.
> > >
> > > # Question 5
> > >
> > > I thank the authors for clarifying their arguments in a revised version of the paper.
> > > I think I now understand what they mean by need of *convergence at any convergence rate*, but I didn't understand it from the reasoning of the paragraph **Intuitive explanation about the influence of the influence of the variance estimation.**
> > >
> > > If we assume that the arm $a$ is pulled $\tilde{w}_a T$ number of times, can't we just use the probability inequality after equation (17) (page 18) in Kaufmann et al. (2016) ? Obviously, we would need some independence between the drawn samples and the estimate of $\tilde{w}_a$ to make it rigorous, but I believe this can be done either using a martingale argument, or by using independent samples for mean estimation and for variance estimation (eg forced exploration samples for variance estimation).
> > >
> > > From there, we can see that to bound the probability, we actually need to bound $\mathbb{E}[\exp(-(\frac{\sigma_1^2}{\tilde{w}_1}+\frac{\sigma_2^2}{\tilde{w}_2})^{-1}T\frac{\Delta^2}{2})]$. From here, it is easier to see that we need the bias of this expectation to not grow at an exponential rate, and that it is not obvious. It seems to me that this bias is actually possible to bound, e.g. using concentration inequalities on chi squared variables, but this would require careful computation and might be out of scope.
> > >
> > > To summarize, I don't ask the reviewers to answer the latter point, but still want the following adjustments in the current paper version:
> > > - make the algorithm and its proof correct and clear
> > > - avoid the need of a lower (or upper, or both) bound on the variances
> > > - make all the big O notations rigorous (see Remark of Reviewer JxB7)
> > >
> > > Also, I think that the claims could be enforced if they were confirmed on a numerical experiment

---

> ### Author Response · Authors · 2024-02-20
> **Reply to Reviewer jcHp**
>
> We are grateful for the reviewer's prompt feedback and constructive insights.
>
> To begin with, we are currently conducting an experiment but have updated the draft to initiate discussion. We will add the experimental results shortly.
>
> ## Question 1
> > Indeed, I actually missed the definition of the thre function at the end of Section 1. Now, I am wondering if there is not a typo in its definition. Following your claim, we should take $C_{\sigma^2}$ arbitrarily small. But in that case, looking at the thre function and the empirical variance estimator, the estimator should actually converge to $1/C_{\sigma^2}$ almost surely. In the light of this, I again reaffirm the need for a rigorous proof of Lemma 5.1. Actually, this observation makes me even doubt on the empirical success of such algorithm. I would thus now join the other reviewers on the need for some numerical experiment.
>
>
> ## Answer 1
> First, we acknowledge the typographical error in the "thre" function's definition and have addressed it in the latest draft. Despite the correction, it seems the reviewer's concern extends beyond this typo. However, we might not get the point of the reviewer's comment that "the estimator should actually converge to $1/C_{\sigma^2}$ almost surely." If the reviewer's comment implies uncertainty about convergence, we argue that such concerns are not the case. Given that each arm is pulled with a sufficiently small positive probability, independent of $T$, it allows for the accumulation of infinite samples for each arm, ensuring convergence. If our understanding your comment is not correct, could you clarify it more?
>
> Moreover, we recognize the practical benefits of forced sampling, as highlighted in the reviewer's second question, in stabilizing empirical performance. This approach is briefly noted at the bottom of page 4, and while it does not impact our theoretical results, it significantly enhances strategy performance. See our Answer 2 in this post.
>
> ## Question 2
> > Even if Lemma 5.1 was correct in its current form, I still think that using a forced exploration of order $\sqrt{T}$ would not change anything about the current analysis, while making it available without any knowledge of a lower (or upper) bound of the variances.
>
> ## Answer 2
> We appreciate this suggestion and concur that forced exploration of order $\sqrt{T}$ ensures a stable sampling probability. Nevertheless, the introduction of a constant is crucial to prevent divergence in the AIPW estimator's $\frac{1}{\widehat{w}_t}$ term. To avoid division by zero, it is essential to truncate $\widehat{w}_t$ with a minimal positive value.
>
> From an empirical standpoint, as the reviewer's comment and also mentioned in our Answer 1, we also encourage the use of $\sqrt{T}$ forced sampling to enhance performance stability. Our initial focus was on the theoretical contributions, aiming to minimize confusion between empirical success and theoretical analysis, which is why it was only briefly mentioned previously. In our next update, we will include experimental results and provide a clear discussion on the benefits of $\sqrt{T}$ forced sampling.
>
> In summary, the application of $\sqrt{T}$ forced sampling should be considered from both theoretical and empirical perspectives. While we acknowledge the empirical advantages of this approach, completely removing the assumption of known $C_{\sigma^2}$ presents challenges due to its role in preventing division by zero.

---

> > ### Comment · Reviewer_jcHp · 2024-02-20
> >
> > Thanks for your quick answer.
> >
> > > First, we acknowledge the typographical error in the "thre" function's definition and have addressed it in the latest draft. Despite the correction, it seems the reviewer's concern extends beyond this typo.
> >
> > I am confused.
> > In the current version, I see the definition of thre as $thre(v; c) = \max(\min(v, c), 1/c)$ for $c\geq 1$. For $c<1$, it is indeed constant to $1/c$.
> >
> > Moreover, the std estimate is defined as $\hat{\sigma}=thre(\tilde{\sigma}, C_{\sigma^2}^{1/2})$. If we pull the arm infinitely often, $\tilde{\sigma}\to\sigma$. Then, we can actually see that if $C_{\sigma^2}^{1/2}\leq \sigma$,  $\tilde{\sigma}\to \max(C_{\sigma^2}^{1/2},1/C_{\sigma^2}^{1/2})$, which is why I thought we would need $C_{\sigma^2}^{1/2}\geq\sigma$.
> >
> > If now we take $C_{\sigma^2}^{1/2}\geq\sigma$, we would have $\tilde{\sigma}\to \max(\sigma,1/C_{\sigma^2}^{1/2})$, and so we would also need $1/C_{\sigma^2}^{1/2}\leq \sigma$ to ensure Lemma 5.1.
> >
> > Overall, it seems that the thre function is needed to clip $\tilde\sigma$ in some bounded interval that excludes $0$. Necessarily, this would require the knowledge of a lower **and** an upper bound of $\sigma$ to ensure that $\sigma$ is indeed in this interval.

---

> ### Author Response · Authors · 2024-02-20
> **Reply to Reviewer jcHp (Cont.)**
>
> > In the current version, I see the definition of thre as $thre(v; c) = \max(\min(v, c), 1/c)$ for $c\geq 1$. For $c<1$, it is indeed constant to $1/c$.
>
> We sincerely apologize for sending our response while we were still writing it. We have updated the draft now.
>
> > I am confused. In the current version, I see the definition of thre as $thre(v; c) = \max(\min(v, c), 1/c)$.
>
> In the previous draft, we flipped the $\max$ and $\min$ inversely. Now, we fix it as $thre(v; c_1, c_2) = \min(\max(v, c_1), c_2)$ by using two values $c_1$ and $c_2$ for clarity.
>
> ## Question 5
> > I thank the authors for clarifying their arguments in a revised version of the paper. I think I now understand what they mean by need of convergence at any convergence rate, but I didn't understand it from the reasoning of the paragraph Intuitive explanation about the influence of the influence of the variance estimation. If we assume that the arm $a$ is pulled $\tilde{w}_a T$
>  number of times, can't we just use the probability inequality after equation (17) (page 18) in Kaufmann et al. (2016) ? Obviously, we would need some independence between the drawn samples and the estimate of $\tilde{w}_a$ to make it rigorous, but I believe this can be done either using a martingale argument, or by using independent samples for mean estimation and for variance estimation (eg forced exploration samples for variance estimation).
>
> > From there, we can see that to bound the probability, we actually need to bound $\mathbb{E}[\exp(-(\frac{\sigma_1^2}{\tilde{w}_1}+\frac{\sigma_2^2}{\tilde{w}_2})^{-1}T\frac{\Delta^2}{2})]$. From here, it is easier to see that we need the bias of this expectation to not grow at an exponential rate, and that it is not obvious. It seems to me that this bias is actually possible to bound, e.g. using concentration inequalities on chi squared variables, but this would require careful computation and might be out of scope.
>
> ## Answer 5
> We are thankful for the reviewer's acknowledgment of our clarifications and for their thoughtful suggestion. At the beginning of our research, we indeed considered a similar approach to that proposed by the reviewer. However, we encountered significant challenges that led us to present the current draft as a noteworthy outcome.
>
> Initially, we attempted to start from the probability inequality subsequent to equation (17) in Kaufmann et al. (2016), but faced obstacles. As a critical challenge, when we estimate the variances, the deterministic variable $A_t$ (and $n_1$ and $n_2$) in Kaufmann et al.'s inequality becomes a random variable, precluding a straightforward application of their analysis.
>
> Despite these hurdles, we acknowledge that if we do not care about the lower bound derived by Kaufmann et al. (2016), we may stil derive an upper bound without the small-gap regime, which might be
> $$\mathbb{E}[\exp(-(\frac{\sigma_1^2}{\tilde{w}_1}+\frac{\sigma_2^2}{\tilde{w}_2})^{-1}T\frac{\Delta^2}{2})] \cdot \mathrm{EstErr},$$
> where $\mathrm{EstErr}$ denotes a term related to the variance estimation error. While the first term corresponds to the lower bound, the EstErr term prohibits us from attaining the matching upper bound. In our previous response, we answer that the EstErr term should be any exponential rate for deriving a matching upper bound.
>
> However, showing such results requires some different technical analysis, and the optimal strategy might not be the Neyman allocation (allocating each arm with the ratio of the standard deviation). We need to change the lower bound. For example, Jourdan et al. (2023) discusses a strategy in a case with unknown variances in the fixed-confidence BAI problem. This would help the next step of our research. In this study, we focused on the optimality of the Neyman allocation since (to the best of our knowledge) it is known to be a difficult problem that requires a specific technique in the existing literature, including van der Laan (2008) and Hahn et al. (2011).
>
>
> ## Question 6
> > make all the big O notations rigorous (see Remark of Reviewer JxB7)
>
> ## Answer 6
> We appreciate your suggestion. Following Reviewer JxB7's remarks, we have fixed big O notation. Specifically, we changed the big O notation to small o notation to clarify what asymptotics we consider and what are constants. We are rigorously reviewing these revision now and will reply to Reviewer JxB7 after that.
>
> -------
>
> We believe that we share the basic ideas with the reviewer, such as the $\sqrt{T}$ forced sampling, the tracking strategy, and the use of Kaufmann et al.'s analysis. In fact, we have already attempted the reviewer's suggestions about the proof strategy using Kaufmann et al.'s results or the tracking strategy in the initial phase of our study but failed. We chose not to include speculative accounts of these attempts in our manuscript, adhering to academic standards. However, we are prepared to discuss our analytical experiences in this rebuttal and are open to further inquiries.

---

> > ### Comment · Reviewer_jcHp · 2024-02-21
> >
> > Thank you for all the detailed answers.
> >
> > >  In fact, we have already attempted the reviewer's suggestions about the proof strategy using Kaufmann et al.'s results or the tracking strategy in the initial phase of our study but failed. We chose not to include speculative accounts of these attempts in our manuscript, adhering to academic standards. However, we are prepared to discuss our analytical experiences in this rebuttal and are open to further inquiries.
> >
> > I think it would be nice to at least explain quickly why such an approach might not work (or at the expense of an extensive technical analysis), as it seems the natural way to tackle the problem. I don't think there is need to provide all the technical details, but only the outline.

---

> ### Author Response · Authors · 2024-02-21
> **Reply to Reviewer jcHp**
>
> Thank you for your interest in this issue. We are eager to address your queries. To answer the reviewer's question, we would like to clarify the reviewer's question and interests. For example, could the reviewer please specify what "such an approach" means? Below, we have outlined what we presume the reviewer might be referring to:
>
> - **Tracking strategy**: The details regarding this have been added to the newly included Section 6.4 in response to the reviewer's query. Please refer to that section for more information.
> - **Strategy using the sample average estimator**: The details regarding this have been added to the newly included Section 6.3 in response to the reviewer's query. Please refer to that section for more information.
> - **Proof using the probability inequality after equation (17) (page 18) in Kaufmann et al. (2016)**: This issue is related to the "Strategy using the sample average estimator." Please see that section first. Similar explanations have been provided in a previous response. Specifically, the upper bound ($\mathbb{E}[\exp(-(\frac{\sigma_1^2}{\tilde{w}_1}+\frac{\sigma_2^2}{\tilde{w}_2})^{-1}T\frac{\Delta^2}{2})] \cdot \mathrm{EstErr}$) in the previous response does not match Kaufmann's lower bound unless $EstErr = exp(-T\epsilon)$ for any $\epsilon > 0$.
>
> Regarding "The probability inequality after equation (17) (page 18) in Kaufmann et al. (2016)" more directly, the proof does not hold when $A_t$ is a random variable. Specifically, the last inequality on page 39 is not valid; that is,
>
> $$ \mathbb{P}(\frac{1}{n_1}\sum^{n_1}X_{1, t} - \frac{1}{n_2}\sum^{n_2}X_{2, t} < 0) $$
>
> $$\leq \mathbb{E}_v[\exp(\lambda \alpha X_{2, t})]^{(1-\alpha) n}\mathbb{E}_v[\exp(\lambda ( 1 - \alpha) X_{1, t})]^{\alpha n} $$
>
> does not hold when $A_t$ is a random variable (we cannot render the above display in a clear way due to the markdown issue). Note that when $A_t$ is a random variable, $n_1$ and $n_2$ are also random variables in Kaufmann et al. (2016), which prevents us from using the proof strategy in Kaufmann et al. (2016).
>
>
> Furthermore, is the reviewer interested in how altering the algorithm or other proofs cannot circumvent this issue, assuming that we accept that the proof of Kaufmann et al. does not hold when $A_t$ is a random variable? In fact, we attempted to circumvent this issue in various ways but failed, including changing strategies. Would it be helpful to explain previous our proof attempts or strategy adjustments that were unsuccessful?
>
> The above replies are intended to provide preliminary answers to the reviewer's questions and to clarify the intent further. If our interpretations do not align with the reviewers inquiries, please ask for further clarification.

---

> > ### Comment · Reviewer_jcHp · 2024-02-22
> >
> > I mostly referred to the point **Proof using the probability inequality after equation (17) (page 18) in Kaufmann et al. (2016)**, but mentioning the other points might also be nice. For your last question, I think it would be enough to claim that making the number of samples random invalidates the inequality of Kaufmann et al. and that it is actually not easy to circumvent this point

---

> ### Author Response · Authors · 2024-02-23
> **Response to Reviewer jcHp**
>
> We would like to express our gratitude for your inquiry. Then, let us address the concerns regarding the "Proof using the probability inequality after Equation (17) on Page 18 in Kaufmann et al. (2016)."
>
> To clarify the issue, let us organize our discussion. The reviewer's point is as follows:
> > Statement 1: It is possible to apply the 'Proof using the probability inequality after Equation (17) on Page 18 in Kaufmann et al. (2016)' to derive an upper bound, even in scenarios where variances are unknown.
>
> Our response is outlined below:
> > Claim 1: Statement 1 does not hold. First, the proof of the ``inequality after equation (17) (page 18) in Kaufmann et al. (2016)'' does not hold when $A_t$ is a random variable. We at least need to change the proof or the definition of the strategy.
>
> For example, as we explained in the previous response, in the proof of ``inequality after equation (17) (page 18) in Kaufmann et al. (2016)'' shown on page 39 in Kaufmann et al. (2016), the following inequality does not hold:
> $$ \mathbb{P}(\frac{1}{n_1}\sum^{n_1}X_{1, t} - \frac{1}{n_2}\sum^{n_2}X_{2, t} < 0) $$
>
> $$\leq \mathbb{E}_v[\exp(\lambda \alpha X_{2, t})]^{(1-\alpha) n}\mathbb{E}_v[\exp(\lambda ( 1 - \alpha) X_{1, t})]^{\alpha n} $$
>
> when $A_t$ is a random variable (we cannot render the above display in a clear way due to the markdown issue). Let us denote the inequality by (*).
>
> To further clarify the point of the problem, we propose the following inquiries to the reviewer:
> - Question 1: Does the reviewer agree or disagree with our Claim 1?
> - Question 2: If the reviewer agrees (Claim 1 is true), could the reviewer suggest a method to validate the inequality (*) when $A_t$ is a stochastic variable?
> - Question 3: If the reviewer disagrees, does the reviewer believe that the original proof in Kaufmann et al. (2016) is sufficient, and the inequality inherently holds, even when $A_t$ is a random variable; that is, in a case where we need to estimate the variances?
> - Question 4: If the reviewer neither agrees nor disagrees with our Claim 1, could the reviewer kindly elaborate on your stance?
>
> If the reviewer agrees with Claim 1, then let us discuss how we show the inequality in Kaufmann et al. (2016) in a case with unknown variances. Through this discussion, we would like to confirm the difficulty. Then, we would elucidate the rationale behind why we take the current approach in our draft to show a similar inequality and why the $o(\Delta^2)$ remains in our derived inequality.
>
> Conversely, if the reviewer believes that the original proof in Kaufmann et al. (2016) holds and does not require modification when $A_t$ is a random variable, or variances are unknown, we would appreciate the opportunity to explain why this proof may not be applicable when $A_t$ a random variable or variances are unknown.

---

### Review · Reviewer_SJ9f · 2024-02-12

**Summary Of Contributions:**

The authors consider the problem of best-arm identification in a two-armed Gaussian bandit with unknown variances. They focus on the fixed budget setting, a less studied setting in comparison with the fixed confidence setting, and thus not fully understood despite its relevance and practical significance. The authors propose a new algorithm, referred to as NA-AIPW, whose probability of mis-identification matches asymptotically, as $T$ the fixed budget tends to infinity and optimality gap $\Delta$ tends to zero, the optimal rate
$$
\frac{\Delta^2}{2(\sigma_1 + \sigma_2)^2}
$$
Their algorithm is based on using a Neyman Allocation (NA) to perform sampling, and an Augmented Inverse Probability Weighing (AIPW) estimator to estimate the unknown means. The proof analysis is thorough. To the best of my knowledge this result is novel.

**Audience:**

Yes

**Broader Impact Concerns:**

I do not foresee any concerns on the ethical implication of the work.

**Claims And Evidence:**

Yes

**Requested Changes:**

I have listed below some concerns that I believe should be addressed, though so not critical.
- Typo in the definition of Neyman Allocation $\omega^\star$? I believe it should be $\sigma_1$ instead of $\sigma_1^2$
- Would the rounds of initialization scale with the number of arms? non-asymptotic regime?
- It is well understood that sampling $A_t$ as you do means that you sample $A_t$ according to Bernoulli random variable with probability parameters $\omega_{1,t}$. So, I am not sure it is worth it to put this much level of detail.
- In proposition 1, you write $(\sqrt{\sigma_1^2} + \sqrt{\sigma_2^2})^2$ and in Theorem 1, you write $(\sigma_1 + \sigma_2)^2$. I think it is best to unify notation here. I would suggest just write it as in Theorem 1.
- In section 4, in the second to last paragraph, the authors write "... simplifies a proof for theoretical analysis.". Please reformulate correctly. The authors also write next "we can decomposed an error ..." Please reformulate in a correct English.
- Wouldn't the big O in step 2 hide some dependence on the variances? It also seems according to your final step that there is a hidden dependence on the variances in the big O. I believe this should be added in the statement of the theorem.
- In step 3, when you invoke the dominated convergence theorem, you need $\omega$ to be bounded away from zero, which to my understanding you enforce thanks to the truncation operator. Don't you need for that knowledge of upper bounds on the means and variances? In which case, I believe you need to precise this. Also, is it possible to avoid such need?  Please provide clarification on this. I believe discussing the need for truncation is also needed.

**Strengths And Weaknesses:**

**Strengths**
- The study of the fixed budget setting is interesting and the authors provide a new result on the problem of the two-armed gaussian bandit with unknown variances.
- The analysis and proof techniques are neat and thorough.
- The contribution is well situated in the literature review.

**Weaknesses**
- The results are provided only for gaussian distributions in the two-armed bandit case. It would be interesting to see if the results can be extended beyond that.
- Theorem 1 is only asymptotic. It would be nice if one can provide a non-asymptotic analysis given that the setting is quite restricted. Perhaps the authors could comment on this and highlight the challenges that arise for such analysis.
- There is a lack of explanation of why truncation is needed and whether upper and lower bounds on the unknown parameters are needed.
- There are no experimental results to confirm the theory or to compare with other existing algorithms that might have knowledge of the variances.

---

> ### Author Response · Authors · 2024-02-22
> **Reply to Reviewer SJ9f**
>
> We appreciate the reviewerWe appreciate the reviewer. We revised our draft. Our replies to each question are listed below.
>
> ## Question 1
> > Would the rounds of initialization scale with the number of arms? non-asymptotic regime?
>
> ## Answer 1
> The first goal of the initialization rounds is to ensure that the estimator is well-defined by avoiding division by zero in our mean estimator. The second goal is to enhance the empirical performance of the proposed strategy. Given that the former is a trivial and technical matter, we focus on elucidating the latter and discuss how many initialization rounds are preferable without affecting the theoretical optimality. If the total number of initialization steps is independent of both $t$ and $T$, we can ignore the theoretical influence of the initialization rounds as $t\to\infty$ and $T\to\infty$. The key assumption is $\hat{w}\to w^*$ as $t\to \infty$ almost surely. We can make the number of initialization steps dependent on the number of arms, which does not affect the theoretical result. Additionally, we can make $\hat{w}$ dependent on $t$ if $\hat{w}\to w^*$ as $t\to \infty$ almost surely holds. For example, using $\hat{w}$ defined in our draft, we can use $\tilde{w} = \alpha_t/2 + (1-\alpha_t)\hat{w}$ as an allocation probability if $\alpha_t \in [0, 1]$ approaches zero as $t\to \infty$. This clarification has been added to the revised manuscript.
>
> ## Question 2
> > It is well understood that sampling $A_t$ as you do means that you sample $A_t$ according to Bernoulli random variable with probability parameters $w_t$. So, I am not sure it is worth it to put this much level of detail.
>
> ## Answer 2
> The detailed process outlined in our initial draft was intended to elucidate the stochastic nature of $A_t$, following the manner of existing studies. However, we acknowledge that such specificity is not crucial for our core findings. In response to the reviewer's feedback for enhanced clarity, we revised our draft and defined that $A_t = a$ with probability $\widehat{w}_{a,t}$.
>
>
> ## Question 3
> > Wouldn't the big $O$ in step 2 hide some dependence on the variances? It also seems according to your final step that there is a hidden dependence on the variances in the big $O$. I believe this should be added in the statement of the theorem.
>
>
> ## Answer 3
> We thank the reviewer for their constructive observation. Indeed, the big $O$ notation hides some parameters, whereas our principal results depend on $o(\Delta^2)$, derived from $O(\Delta^3)$. The big $O$ and small $o$ notations denote asymptotic behaviors in $\lambda \to 0$ (or $\Delta\to 0$). In the revised manuscript, we have used $o(\lambda^2)$ directly instead of using $O(\lambda^3)$ and clarified that we consider the asymptotics of $\lambda \to 0$ (or $\Delta\to 0$). In the revised draft, we also aim to clarify the dependencies as possible. However, since the small $o$ notation denotes terms that vanish as $\lambda \to 0$ (or $\Delta\to 0$), we do not mention such hidden parameters so much, provided they remain constant relative to $\lambda \to 0$ (or $\Delta\to 0$). If the reviewer still recommends us to write some dependence in $o$, we will add them more in the next revision.
>
> ## Question 4
> > In step 3, when you invoke the dominated convergence theorem, you need $w$ to be bounded away from zero, which to my understanding you enforce thanks to the truncation operator. Don't you need for that knowledge of upper bounds on the means and variances? In which case, I believe you need to precise this. Also, is it possible to avoid such need? Please provide clarification on this. I believe discussing the need for truncation is also needed.
>
> ## Answer 4
> As the reviewer pointed out, the boundedness $w$ is imposed via our truncation operator, and we need to know the upper bounds of $|\mu_a|$ and the upper and lower bounds of $\sigma^2_a$. We clarified this assumption in the revised draft. Note that we can just set a very small value for the lower bound of $\sigma^2_a$ and a very large value for the upper bounds of $|\mu_a|$ and $\sigma^2_a$.
>
> ## Question (Typos)
> > Typo in the definition of Neyman Allocation $w^*$? I believe it should  be $\sigma_1$ instead of $\sigma^2_1$.
>
> > In proposition 1, you write $(\sqrt{\sigma_1^2} + \sqrt{\sigma_2^2})^2$ and in Theorem 1, you write $(\sigma_1 + \sigma_2)^2$. I think it is best to unify notation here. I would suggest just write it as in Theorem 1.
>
> > In section 4, in the second to last paragraph, the authors write "... simplifies a proof for theoretical analysis.". Please reformulate correctly. The authors also write next "we can decomposed an error ..." Please reformulate in a correct English.
>
> ## Answer
> We are thankful to the reviewer for identifying these typographical errors. They have indeed been rectified in the revised draft.

---

### Review · Reviewer_JxB7 · 2024-02-13

**Summary Of Contributions:**

The authors consider the best arm identification problem with two decisions in the fixed budget setting, where each of the two decisions have a Gaussian reward distribution with both unknown means and variances. The goal is for the learner to sample adaptively one of the two decisions, and after $T$ samples have been collected, recommend the decision with the highest expected reward. The metric to minimize is the probability of reccomending the suboptimal arm.

In this problem when the variances are known, optimal strategies are known, and involve sampling each decision with probability proportional to its variance (this is called the Neyman allocation in the statistical literature). Since in the problem at hand the variances are not known in advance, the authors propose to estimate the variances as the algorithm goes, with some trimming to avoid corner cases in which the probability of sampling any of the two decisions is null. Interestingly, the authors do not consider the obvious recommendation rule where one simply maximizes the empirical mean. Rather, they use a weighted average of the rewards (where the weights are the inverse probabilities of selecting a given decision), which (by their admission) eases the analysis, and also tends to give more weight to decisions that have been selected less often.

Their main technical claim is that this strategy reaches the lower bound derived in the case where the variance is known, in a particular regime where the expected rewards of the two decisions becomes close (i.e the small "gap" regime).

**Audience:**

Yes

**Broader Impact Concerns:**

N.A.

**Claims And Evidence:**

Yes

**Requested Changes:**

See above.

**Strengths And Weaknesses:**

Strengths:

Although the proposed strategy combines several known elements from the litterature, the approach seems novel, and is provably optimal in a well chosen regime. The paper is clearly written, and the analysis seems original.

Weaknesses

- The authors argue that their strategy is only provably optimal in the (narrow) regime where gaps are small, but outside of this regime one cannot a priori assess how good the proposed strategy performs. This is typically the type of question which should be addressed using numerical experiments, and the performance of this strategy should be compared to the many already available strategies of the litterature, as well as a lower bound constituted by the optimal algorithm which knows the variances in advance. Doing this for both small and large gaps would certainly yield insight into this question.

- In "Step 2; Evaluation by using the Chernoff Bound": I do not think that equation (4) is true without an additional argument, namely that the third moment of $\Phi_t$ exists, otherwise the Taylor expansion of the moment generating function is not valid. Furthermore, the notation $O(\lambda^3)$ is probably incorrect, in the sense that there are quantities hiding into the $O(.)$ that are not universal constants, and depend on the problem parameters, indeed the $\hat{w}$ need to be bounded from below, and this bound necessarily depends on the problem parameters. Notice that this carries to the result as well, where there are problem dependent terms that creep into the $O(\Delta^3 )$ term. Overall I feel the analysis should be redone by specifying how all the terms involved depend on the problem parameters.

- Along the same lines, the authors claim that the expectation of ${\Phi_t^k \over k!}$ is bounded by some "constant" (I am not sure that this can be made independent of the problem parameters) that does not depend on $k,t$. Why ? and what is this "constant" ?

- Also, as far as I understand, $\Phi_t | F_{t-1}$ is a mixture of two Gaussians, so at this rate why not compute its moment generating function straight away ?

- Minor remarks: "is aligns" (typo); "In statistical test" (grammar) ; $\sqrt{\sigma_1^2}$ -> $\sigma_1$ (notation);  in 3.1 "Target allocation Ratio" the formula for $w_1^\star$ is incorrect;  the acronym "MDS" is not defined (I am assuming it means Martingale Difference Sequence ...);

---

> ### Author Response · Authors · 2024-02-22
> **Reply to Reviewer JxB7**
>
> We appreciate the reviewer's constructive feedback and have revised the manuscript accordingly. Our responses to your inquiries are outlined below.
>
> -----------
>
> ### Question 1
> > The authors argue that their strategy is only provably optimal in the (narrow) regime where gaps are small, but outside of this regime one cannot a priori assess how good the proposed strategy performs. This is typically the type of question which should be addressed using numerical experiments, and the performance of this strategy should be compared to the many already available strategies of the litterature, as well as a lower bound constituted by the optimal algorithm which knows the variances in advance. Doing this for both small and large gaps would certainly yield insight into this question.
>
> ### Answer 1
> We acknowledge your suggestion and are currently conducting simulation studies to evaluate the empirical performance of our proposed strategy. These results will be incorporated into the next revision in a few days.
>
> -----------
>
> ### Question 2
> In "Step 2; Evaluation by using the Chernoff Bound": I do not think that equation (4) is true without an additional argument, namely that the third moment of $\Psi_t$ exists, otherwise the Taylor expansion of the moment generating function is not valid.
>
> ### Answer 2
> Since $Y_{a,t}$ are Guassian, the other varialbes bounded, and $1/\hat{w}$ is upper bounded from the truncation of $\hat{w}$, the third moment of $\Psi_t$ exists. We clarified this point in the revised draft.
>
> -----------
>
> ### Question 3
> > Furthermore, the notation $O(\lambda^3)$ is probably incorrect, in the sense that there are quantities hiding into the $O(\cdot)$ that are not universal constants, and depend on the problem parameters, indeed the need to be bounded from below, and this bound necessarily depends on the problem parameters. Notice that this carries to the result as well, where there are problem-dependent terms that creep into the $O(\Delta^3)$ term. Overall I feel the analysis should be redone by specifying how all the terms involved depend on the problem parameters.
>
> ### Answer 3
> We appreciate the reviewer's emphasis on clarity regarding the hidden variables within the $O(\cdot)$ notation. Although the $O(\cdot)$ represents asymptotics relative to $\lambda$ (or $\Delta$), we agree that specifying the dependency on other fixed parameters would enhance rigor. The revised manuscript clarifies these dependencies and transitions from $O(\lambda^3)$ to $o(\lambda^2)$ (or $o(\Delta^2)$), following Peano's form of the Taylor remainder. This change obviates the need to specify hidden variables for $o(\lambda^2)$, as their influence diminishes as $\lambda$ approaches zero. Due to this change, we do not mention the hidden variables in $o(\lambda^2)$ because the $o(\lambda^2)$ implies that it approaches zero as $\lambda \to 0$, and the hidden variables do not matter. We hope this adjustment aligns with the reviewer's expectations. If the reviewer still recommends us to write some dependence in $o$, we will add them more in the next revision.

---

> > ### Author Response · Authors · 2024-02-22
> > **Reply to Reviewer JxB7 (Cont.)**
> >
> > ### Question 4
> > > Along the same lines, the authors claim that the expectation of $\Phi^k_t / k!$ is bounded by some "constant" (I am not sure that this can be made independent of the problem parameters) that does not depend on $k,t$. Why ? and what is this "constant" ?
> >
> > ### Answer 4
> > Thank you for pointing out the problem. We believe that the description was confusing due to our definition of bandit models and the lack of explanation. First, we intended to consider Gaussian bandit models with fixed variances but variable means; that is, in the set of distributions, the variances are fixed, while the means take various values. Furthermore, we consider finite means and variances. Second, we truncated $\hat{w}_t$ in $o(\lambda^2)$ using the finiteness of the variances. Note that the variances do not depend on a problem $P$ since the variance is fixed for $\mathcal{P}^{\mathrm{G}}$; that is, while $\mathcal{P}^{\mathrm{G}}$ depends on the choice of the variances, a problem $P\in\mathcal{P}^{\mathrm{G}}$ does not depend on the variances (for all $P$, their variances are the same). For these two points, we guarantee the finiteness of $\mathbb{E}[\Phi^k_t / k!]$.
> >
> > Following your comments, we fixed our description. During the revision, we found that we do not have to assume the boundedness of $\mathbb{E}[\Phi^k_t / k!]$ for all $k$, but it is enough to check the existence of the first, second, and third derivatives of $\mathbb{E}\left[\exp(\lambda \Psi_t)\right]$ w.r.t. $\lambda\in(0, \infty)$ (we can make more rigorous explanation, but we believe that this statement is enough for proof). We corrected the proof by checking the conditions.
> >
> > ---------
> >
> > ### Question 5
> > > Also, as far as I understand, $\Psi_t | \mathcal{F}_{t-1}$ is a mixture of two Gaussians, so at this rate why not compute its moment generating function straight away?
> >
> > ### Answer 5
> > As the reviewer pointed out, $\Psi_t | F_{t-1}$ is a mixture of two Gaussians. However, we think that it does not help our analysis so much. First, the moment-generating function of a mixture of two Gaussians does not have a simple form (note that due to the randomness of $A_t$, we cannot use the properties of moment-generating functions of Gaussian random variables). Second, we need to take the expectation over $F_{t-1}$ at later steps, which also makes the analysis complex. If $A_t$ is deterministic, we can separately obtain simplified forms of moment-generating functions of two Gaussians, but the randomness of $A_t$ prevents us from using it. Due to the above reasons, we did not take the approach.
> >
> > ---------
> >
> > ### Question 6
> > > Typos...
> >
> > ### Answer 6
> > We are grateful for the reviewer's attentiveness to detail and have corrected the typographical errors accordingly.

---

### Author Response · Authors · 2024-02-23
**Revision**

Dear Reviewers,

We appreciate the reviewers for your insightful and constructive feedback. Following your suggestions, we have revised our manuscript. The major changes include (i) the collection of the big O-notation and (ii) the incorporation of a new Section 6, which delves into the complexities associated with this problem.

We are in the process of adding experimental results and will revise the draft again in a few days. Prior to incorporating these results, we updated the draft to start the discussion with the reviewers.

Best,
Authors

---

> ### Author Response · Authors · 2024-02-26
> **On Experimental Results**
>
> Dear AE and Reviewers,
>
> We are grateful for your constructive feedback. In response to your comments, we are running experiments to add the results in our draft. However, these are expected to take a few more days, or at most one week, to complete. We kindly request your patience as we finalize these results.
>
> Best regards,
> The Authors

---

> > ### Author Response · Authors · 2024-03-01
> > **Second Revision**
> >
> > Dear Reviewers,
> >
> > We apologize for the delay in the revision of our manuscript. We encountered conflicts in lab computer usage, which resulted in server unavailability and subsequently delayed our experimental results. We have updated the manuscript with the results that are currently available. Should we manage to obtain additional experimental results, we will further update the manuscript. We will also continue to correct any typos. For now, the major revisions have been completed.
> >
> > Regards,
> > Authors

---

### Decision · Action_Editor_Amrd · 2024-03-11

**Recommendation:** Reject

**Comment:**

While the reviewers agree that the paper has the potential to be published, some fixes are still needed before the paper can be accepted. Since these fixes will require a second round of review, which is not available when accepting with minor revision, the submission is rejected. Nonetheless, the authors are encouraged to make the necessary corrections detailed under "Claims and Evidence" and resubmit.
Please also review again all of the reviewer comments and make any additional corrections that may improve the readability and accuracy of the derivations. Optionally, you may also wish to address the following post-discussion comment:

The algorithm now needs knowledge of upper/lower bounds on unknown parameters such as the means and variances. It would be more satisfactory to avoid such limitation, especially since the setting is so simple (two-armed gaussian bandits).
If you are unable to achieve that, it would be helpful to explain to the reader the main hurdles.

**Audience:**

This paper studies a best arm identification problem with two arms in the fixed budget setting, where each of the two arms have a Gaussian reward distribution with both unknown means and variances. An algorithm is proposed that is optimal in the small-gap regime. The result would be of interest to the relevant TMLR audience.

**Claims And Evidence:**

The reviewers have identified several issues regarding the accuracy of the claims and analysis. While many of the issues have been addressed by the authors, there remains a widespread issue that would require a second reviewing round after correcting. This issue has been detailed in a post-discussion review comment:

After the authors's rebuttal, there is now an additional dependence in \Delta in the main result (new term o(\Delta^2)). In the final step it seems the authors have o(\Delta^2/V) but in the main statement the dependence on V is removed. I would ask the authors to precise all constant dependencies and actually avoid any small o notations if the paper is to be accepted. Using such notations makes things very confusing and not so rigorous (other reviewers also expressed this concern) especially in clarifying in what sense optimality is attained.

**Resubmission Of Major Revision:**

The authors may consider submitting a major revision at a later time.

---

> ### Author Response · Authors · 2024-03-17
> **Thank you for reviewing our manuscript**
>
> Dear Action Editor and Reviewers,
>
> Thank you for reviewing our manuscript. We appreciate the feedback provided and are committed to making sincere revisions for resubmission. However, we would like to address one point for clarification, as it is pertinent to our discussion, especially considering its presence on OpenReview.
>
> Our paper concerns the optimality as $\Delta \to 0$. The term $o(\Delta^2)$ in our work does not depend on $V$ mathematically. Indeed, for the performance metric, denoted by $f(\Delta)$ here for simplicity, we state $f(\Delta) = o(\Delta^2/V) = o(\Delta^2)$. This is because our focus lies on the asymptotics with respect to $\Delta$, not $V$. Here, the usage of Little-o notation $o(\Delta^2)$ implies that
> > for every positive constant $\epsilon > 0$, there exists a constant $\tilde{\Delta} > 0$ such that for all $\Delta\geq \tilde{\Delta}$, $f(\Delta)\leq \epsilon\Delta^2$.
>
> Since $V$ is included in $\epsilon$ (we can just define $\epsilon = \epsilon' V$ for every positive $\epsilon' > 0$ since $V$ is fixed regarding $\Delta \to 0$), introducing a fixed constant $V$ alongside $\Delta$ would be a mathematically unnatural operation from the viewpoint of the little-o notation. It is more common or mathematically appropriate to remove $V$. Thus, for the purposes of asymptotic analysis with respect to $\Delta$, we believe our usage of Little-o notation is precise and rigorous. For further clarification, for example, please see the Wikipedia article on Little-o notation at \url{https://en.wikipedia.org/wiki/Big_O_notation#Little-o_notation}.
>
> The use of little-o notation is also common in asymptotic analysis. For example, in various studies including Kaufmann et al. (2016), Garivier et al. (2016), Komiyama et al. (2022), Komiyama et al. (2023) and Degenne (2023), they use $lim_x f(x) = C$ or $f(x) = C + o(1)$ as $x\to 0$, where $f(x)$ is some metric that we want to evaluate depending on some value $x$, where both $\lim_x f(x) = C$ and ``$f(x) = C + o(1)$ as $x\to 0$'' have the mathematically same meaning.
>
> - Kaufmann, E., O. Capp ́e, and A. Garivier (2016): “On the Complexity of Best-Arm Identification in Multi-Armed Bandit Models,” Journal of Machine Learning Research, 17(1), 1–42.
> - Garivier, A., and E. Kaufmann (2016): “Optimal Best Arm Identification with Fixed Confidence,” in Conference on Learning Theory.
> - Komiyama, J., Ariu, K, Kato, M., and Qin, C (2023): ``Rate-Optimal Bayesian Simple Regret in Best Arm Identification,'' Mathematical Operations Research.
> - Rémy Degenne. On the existence of a complexity in fixed budget bandit identification. In Conference on Learning Theory, volume 195, pp. 1131–1154. PMLR, 2023.
> - Komiyama, J, Tsuchiya, T, and Honda, J (2022): ``Minimax optimal algorithms for fixed-budget best arm identification.'' In Advances in Neural Information Processing Systems.
>
> Since our main claim is that the upper and lower bounds match as $\Delta \to 0$, what we want to show is $\lim \inf_{T\to\infty} - \frac{1}{T}\log P \geq \frac{\Delta^2}{(\sigma^1 + \sigma^2)^2} - o(\Delta^2)$ as $\Delta \to 0$, or equivalently $\lim \inf_{\Delta \to 0}\lim \inf_{T\to\infty} - \frac{1}{\Delta^2 T}\log P \geq \frac{1}{(\sigma^1 + \sigma^2)^2}$, which is shown in the current draft. That is, it is enough or more mathematically appropriate to use $o(\Delta^2)$ (note again that $o(\Delta^2)$ does not depend on $V$ and other parameters in the asymptotics of $\Delta$). This usage also aligns with existing studies. If we use $o(\Delta^2 / V)$, it alters the meaning of the result since it represents the asymptotics of $\Delta / \sqrt{V} \to 0$, which is not considered in our study.
>
> Thank you once again for your valuable feedback, and we look forward to your guidance on the above problem.
>
> Sincerely,
> Authors
>
> (Additionally, regarding the algorithm's reliance on upper/lower bounds of unknown parameters such as means and variances, we have provided some explanation in the manuscript. However, we aim to enhance the clarity of this aspect further to improve readability)

---

> > ### Comment · Action_Editor_Amrd · 2024-03-27
> > **Asymptotics**
> >
> > Dear Authors,
> >
> > It is agreed that your use of o($\Delta^2$) does describe the correct asymptotics.
> > Nonetheless, there is some ambiguity about your claim that the upper bound and lower bound are matching in the small gap regime. The difference between the upper bound and the lower bound is o($\Delta^2$). While this quantity vanishes when $\Delta\to\infty$, this is not very helpful, since the bound itself also vanishes in this regime. Your claim of approaching optimality may be more justifiable when considering the ratio between the bounds, which approaches 1 as $\Delta\to\infty$. In your revision, please refine your claims of optimality to address this point.

---

> ### Author Response · Authors · 2024-04-01
> **Response to AE**
>
> Dear AE,
>
> Thank you for your response. We have carefully revised our manuscript according to your valuable comments and resubmitted it. Should we designate you as the Action Editor again? If so, could you kindly make yourself "Available" for the assignment? We apologize for the inconvenience and deeply appreciate your cooperation.
>
> Furthermore, we would like to address the following comment:
> > It is agreed that your use of $o(\Delta^2)$ does describe the correct asymptotics. Nonetheless, there is some ambiguity about your claim that the upper bound and lower bound are matching in the small gap regime. The difference between the upper bound and the lower bound is $o(\Delta^2)$. While this quantity vanishes when $\Delta\to \infty$, this is not very helpful, since the bound itself also vanishes in this regime. Your claim of approaching optimality may be more justifiable when considering the ratio between the bounds, which approaches 1 as $\Delta\to \infty$. In your revision, please refine your claims of optimality to address this point.
>
> In response to this comment, we would like to clarify the following points:
> 1. Our asymptotic analysis is as $\Delta \to 0$ (or equivalently, $1/\Delta \to \infty$).
> 2. While the term $o(\Delta^2)$ vanishes as $\Delta\to \infty$, it does not imply that the bound itself vanishes.
>
> Regarding the second point, mathematically, $o(\Delta^2)$ implies that that for a function $f(\Delta)$ and for any $\epsilon > 0$, there exists $\delta(\epsilon)> 0$  such that for all $0 < \Delta < \delta(\epsilon)$, it holds that $f(\Delta)\leq \epsilon \Delta^2$. For instance, $\Delta^2 + o(\Delta^2)$, mathematically, means $\Delta^2 + \epsilon \Delta^2$ for any $\epsilon$. This notation is standard and widely used in statistics. For reference, see page 134 in Durret's "Probability: Theory and Examples" (5th edition, 2019) and page 154 in Billingsley's "Probability and Measure" (3rd edition, 1995), which are globally used textbooks. In our revised manuscript, we have included several equivalent formulations, including those using ratios as the AE suggested, for the convenience of our readers.
>
> Durret, "Probability: Theory and Examples" (v5), 2019.
> Billingsley, "Probability and Measure" (v3), 1995.